# Learning Compact Representations of LLM Abilities via Item Response Theory

## Abstract

Recent years have witnessed a surge in the number of large language models (LLMs), yet efficiently managing and utilizing these vast resources remains a significant challenge. In this work, we explore how to learn compact representations of LLM abilities that can facilitate downstream tasks, such as model routing and benchmark prediction. We frame this problem as estimating the probability that a given model will correctly answer a specific query. Inspired by the item response theory (IRT) in psychometrics, we model this probability as a function of three key factors: (i) the model's multi-skill ability embedding $\theta$, (ii) the query's discrimination vector $\alpha$ that separates models of differing skills, and (iii) the query's difficulty scalar $\beta$. To learn these parameters jointly, we introduce a Mixture-of-Experts (MoE) network that couples model- and query-level embeddings. Extensive experiments demonstrate that our approach leads to state-of-the-art performance in both model routing and benchmark accuracy prediction. Moreover, analysis validates that the learned parameters encode meaningful, interpretable information about model capabilities and query characteristics.

## 1 Introduction

Recent years have seen an explosion of large language models (LLMs), spanning from massive, general-purpose systems to small, task-specialized ones. As of August 2025, Hugging Face lists over 97,000 text-generation models with at least 6B parameters. This proliferation, however, introduces a significant challenge: how to *efficiently* manage and utilize such a vast, rapidly expanding ecosystem.

A crucial step toward addressing this challenge is to construct *compact* representations of models' abilities (Zhuang et al., 2025). By encoding each model's strengths and weaknesses, such representations can facilitate a range of downstream applications. For example, in *model routing*, they allow for assessing a model's suitability for a given query, so that queries can be assigned to the most appropriate model within a candidate pool (Shnitzer et al., 2023; Lu et al., 2024; Jitkrittum et al., 2025). This not only helps balance performance and cost (Ong et al., 2024; Feng et al., 2024; Wang et al., 2025; Frick et al., 2025), but also empowers ensembles of smaller models to effectively compete with large proprietary systems (Zhang et al., 2025e; Pan et al., 2025). Another use case is *benchmark prediction* (Polo et al., 2024b; Zhang et al., 2025a). Traditionally, benchmarks are designed to compare the abilities of different models on specific tasks. However, the recent surge in their number [1] makes it impractical for exhaustive evaluation. Compact representations of model abilities offer an alternative, enabling efficient, scalable LLM evaluation.

In this work, we propose a novel approach to construct compact representations of model abilities as shown in Figure 1, drawing inspiration from the item response theory (IRT). IRT is a well-established statistical framework used in education and psychology to measure latent abilities through standardized tests. It models the interactions between individuals' performance, their latent abilities, and query characteristics, such as difficulty and discrimination (as in the 2-parameter IRT). By treating queries as tests and LLMs as test respondents, we analogously model the likelihood of an LLM $m$ correctly answering a query $q$ as a mathematical function of $m$'s latent abilities $\theta_m$, query

---

[1] For example, NeurIPS 2025's dataset and benchmark track received more than 1,900 submissions, and in 2024, more than 1,200 submissions; a large proportion of accepted submissions are LLM benchmarks.

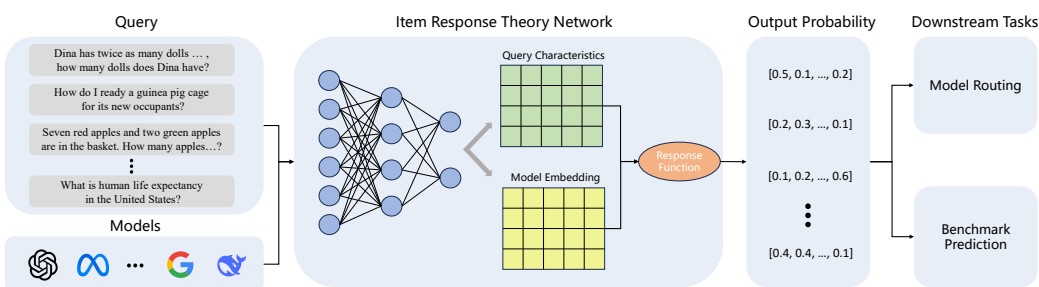

Figure 1: Overview of the IrtNet framework for learning LLM representations. IrtNet learns model embeddings based on models' past query answering performance and outputs probabilities that models answer correctly. The output probability can be directly applied to downstream tasks containing model routing and benchmark prediction.

$q$'s difficulty $\beta_q$, and $q$'s discrimination power $\alpha_q$ between models. To estimate the model's compact representation of abilities ($\theta_m$) and the query characteristics ($\beta_q$ and $\alpha_q$), we present a neural network *IrtNet*. This architecture enables end-to-end training, allowing for the simultaneous optimization of model latent abilities and query characteristics to align with the ground truth of whether a given model correctly answers a given query.

Comprehensive experiments show that our approach not only achieves state-of-the-art performance on downstream tasks but also produces interpretable learned representations. Specifically, in model routing, our method achieves an average accuracy of 67.4%, significantly outperforming strong baselines like RouterDC (Chen et al., 2024) (54.9%), MODEL-SAT (Zhang et al., 2025c) (56.7%), Avengers-Pro (Zhang et al., 2025e) (62.1%), EmbedLLM (Zhuang et al., 2025) (60.2%). In benchmark prediction, our method demonstrates remarkable data efficiency. It reaches 69.9% accuracy using less than 4% of the training data, matching the state-of-the-art (EmbedLLM) performance achieved with the full training set. Moreover, our analysis validates that the learned parameters encode meaningful information: models' compact representations form clear clusters in the latent space based on model family (e.g., Llama, Qwen series) and specialization (e.g., models trained for coding and mathematics); the learned query difficulty parameter $\beta_q$ exhibits a near-perfect negative correlation with true benchmark scores, with a Pearson correlation coefficient of -0.9721. These findings indicate that IrtNet successfully captures the intrinsic model abilities and query characteristics, providing a powerful new tool for evaluation, selection, and management of the vast, rapidly expanding LLM ecosystem.

## 2 PRELIMINARY

We denote the compact representation of an LLM $m$ by a $d$-dimensional embedding vector $\theta_m \in \mathbb{R}^d$, which encodes the model's strengths and weaknesses. Let $M = \{m_1, m_2, \ldots, m_n\}$ be a set of $n$ distinct LLMs, and let $Q = \{q_1, q_2, \ldots, q_k\}$ be a set of $k$ distinct queries. For any given model-query pair $(m, q)$, where $m \in M$ and $q \in Q$, the model's answer to the query can be represented by a binary outcome $y \in \{0, 1\}$, where $y = 1$ signifies a correct answer and $y = 0$ signifies an incorrect one.

Our objective is to learn a probability mass function $f_\theta(m, q) = \Pr(y = 1|m, q)$, parameterized by $\theta = (\theta_1, \ldots, \theta_n)$, such that for any given model-query pair, the function $f_\theta$ can accurately predict whether the model can answer the query correctly or not. Once learned, the function $f_\theta$ can facilitate several important downstream tasks:

**Model Routing.** Given a set of candidate models, model routing aims to assign each query to the most suitable model, avoiding the need for every model to answer it exhaustively Shnitzer et al. (2023). In this way, the ensemble of models effectively leverages collective intelligence, allowing the group to solve a broader range of tasks that any individual model alone cannot accomplish. Formally, let $\mathcal{M} \subseteq M$ be the set of candidate models; based on the function $f_\theta$, for any query $q$, model routing can be achieved by selecting the model $m^*$ that yields the highest probability of

generating a correct answer, i.e.,

$$m^* = \arg\max_{m \in \mathcal{M}} f_\theta(m, q). \tag{1}$$

**Benchmark Prediction.** As exhaustive LLM evaluation is compute-intensive and time-consuming, benchmark prediction seeks to estimate the overall performance of an LLM from only a small subset of evaluation data (Vivek et al., 2023). Formally, let $\mathfrak{Q}$ be the set of queries that are not included when learning the function $f_\theta$, i.e., $\mathfrak{Q} \cap Q = \emptyset$, where typically $|\mathfrak{Q}| \gg |Q|$; for a given model $m$, benchmark prediction generates a predicted accuracy $\hat{S}$ for the set $\mathfrak{Q}$ by feeding each query from this set into the function $f_\theta$, which outputs the predicted probability of generating correct answers to these queries, i.e.,

$$\hat{S} = \frac{1}{|\mathfrak{Q}|} \sum_{q \in \mathfrak{Q}} f_\theta(m, q). \tag{2}$$

In practice, as $f_\theta$ is a neural predictor, a single pass through all queries in the set $\mathfrak{Q}$ can yield the predicted accuracy for all models in the set $M$. This avoids the need to run multiple passes (one pass for each model) as in traditional benchmarking LLMs, thereby allowing for efficient, scalable LLM evaluation.

Overall, establishing compact representations of LLMs provides a unified framework for understanding a model's strengths and weaknesses, supporting a range of downstream applications that have attracted significant recent interest.

## 3 METHODOLOGY

### 3.1 MODELING LLM ABILITIES VIA ITEM RESPONSE THEORY

The item response theory (IRT) is a well-established statistical framework to measure the latent abilities of respondents through testings (Cai et al., 2016). It is widely applied in education and psychology, and underpins the scoring scales of high-stakes exams such as the GRE and GMAT.

The IRT is based on the idea that the probability of a correct response to an item (or a question) is a mathematical function of a person's latent traits, indicating the person's ability, as well as the item parameters. Here, we focus on the two-parameter IRT model, which considers an item's difficulty and discrimination (its ability to differentiate between individuals) (Reise & Waller, 2009).

Understanding LLM abilities through their performance on queries is analogous to assessing a person's abilities based on their performance on standardized tests. Thus, inspired by the IRT, we treat each LLM $m$ as a respondent with a latent trait, and each query $q$ as an item with two parameters that characterize its difficulty and discriminative power.

Formally, let $\alpha_q \in \mathbb{R}^d$ denote the query $q$'s $d$-dimensional discrimination parameter, and $\beta_q \in \mathbb{R}$ denote $q$'s difficulty parameter; we assume that the function $f_\theta(m, q)$, which predicts whether the model $m$ can correctly answer the query $q$, takes the following form of an item response function:

$$f_\theta(m, q) = \sigma\big(\alpha_q^\top \theta_m - \beta_q\big) = \frac{1}{1 + e^{-(\alpha_q^\top \theta_m - \beta_q)}}, \tag{3}$$

where $\theta_m \in \mathbb{R}^d$ is the model $m$'s compact representation (defined in Section 2), indicating its latent abilities, and $\sigma(\cdot)$ is the logistic link function. The discrimination parameter $\alpha_q$ suggests how important each latent ability dimension for the query is, where a higher value indicates that proficiency in that dimension is more critical for a correct answer. The dot product $\alpha_q^\top \theta_m$ computes how well the model's abilities align with the discriminative power of the query. Essentially, it measures the model's fit to the query in terms of both the model's ability in relevant latent dimensions (encoded by $\theta_m$) and the importance of those dimensions to the query (encoded by $\alpha_q$).

### 3.2 IRTNET: LEARNING COMPACT REPRESENTATIONS

We propose to jointly learn models' compact representations $\theta = (\theta_1, \ldots, \theta_n)$, queries' discrimination parameters $\alpha = (\alpha_1, \ldots, \alpha_k)$, and difficulty parameters $\beta = (\beta_1, \ldots, \beta_k)$ through a neural network *IrtNet* in an end-to-end manner.

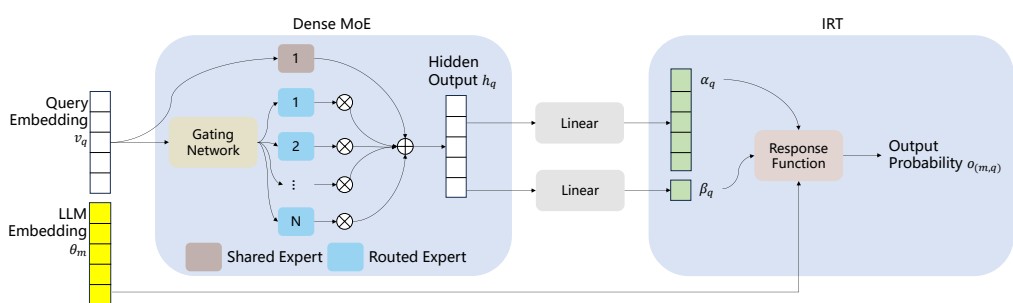

Figure 2: The architecture of IrtNet. A query embedding is processed through a dense MoE layer and subsequent linear layers to generate the query's discrimination $\alpha_q$ and difficulty $\beta_q$ parameters. These parameters are then combined with an LLM embedding $\theta_m$ via the response function to compute the final output probability.

We illustrate the architecture of IrtNet in Figure 2. First, each query $q$ is converted into a semantic embedding $v_q$ using a pre-trained embedding model. Next, $v_q$ is fed into the MoE layer of the IrtNet. We use a dense MoE layer which always activates all $N$ experts, with an auxiliary-loss-free load balancing strategy (Liu et al., 2024) to promote a diverse weight distribution. This design aims to capture diverse, multi-faceted understanding of the query, improving prediction accuracy while avoiding the training instability problem often encountered in sparse MoE layers. The query's hidden output $h_q$ is obtained by

$$h_q = \text{MoE}(v_q) = \text{SharedExpert}(v_q) + \sum_{i=1}^{N} w_i \cdot \text{RoutedExpert}_i(v_q). \tag{4}$$

Next, we employ two independent linear layers to obtain the discrimination parameter $\alpha$ and difficulty parameter $\beta$:

$$\alpha_q = \text{Linear}(h_q, d), \quad \beta_q = \text{Linear}(h_q, 1). \tag{5}$$

where $d$ is the dimension of $\theta_m$. Finally, the IrtNet combines the model embedding $\theta_m$ with the query characteristic parameters $\alpha_q$ and $\beta_q$ to compute the output $o_{(m,q)}$ based on the response function shown in Equation 3.

To learn these parameters, we define the objective as minimizing the discrepancy between the predicted probabilities $o_{(m,q)}$ and the ground-truth labels $y$ across the entire training dataset. Specifically, for a training set $\mathcal{D}$ consisting of samples $(m, q, y)$, the overall loss $\mathcal{L}$ is formulated as the sum of the binary cross-entropy losses for all samples:

$$\mathcal{L} = - \sum_{(m,q,y)\in\mathcal{D}} [y \log(o_{(m,q)}) + (1 - y) \log(1 - o_{(m,q)})]. \tag{6}$$

## 4 EXPERIMENTS

### 4.1 EXPERIMENTAL SETUP

**Datasets** We use the same data as EmbedLLM (Zhuang et al., 2025). Data repository: `https://huggingface.co/datasets/RZ412/EmbedLLM`. We applied a majority vote to consolidate multiple answers from a model to the same query. This step ensures a unique ground truth for each model-query pair, which is especially critical for the test set. The datasets contain 35,673 queries from 10 public benchmarks, including ASDiv (Miao et al., 2020), GPQA (Rein et al., 2024), GSM8K (Cobbe et al., 2021), MathQA (Amini et al., 2019), LogiQA (Liu et al., 2020), MedM-CQA (Pal et al., 2022), MMLU (Hendrycks et al., 2021), SocialIQA (Sap et al., 2019), PIQA (Bisk et al., 2020), and TruthfulQA (Lin et al., 2022). The correctness of answers from 112 open-source language models to those queries was evaluated. The queries were converted into 768-dimensional embeddings using the all-mpnet-base-v2 (Reimers & Gurevych, 2019) sentence transformer. The queries were split into a training set of 29,673 queries, a validation set of 3,000 queries, and a test set of 3,000 queries. LLMs contained are shown in Appendix A.6

**Hyperparameters** The dimension $d$ of the compact LLM representation $\theta_m$ is 232. The number of experts in the MoE layer is set to 40, which is 4 times the number of datasets.

## 4.2 MODEL ROUTING

In this section, we test IrtNet's ability to serve as an effective and intelligent router in a multi-model environment. The objective is to leverage the fine-grained predictions from our framework to select the best model from a diverse pool to handle a given query, thereby maximizing both accuracy and efficiency. For any given query $q$, we compute the predicted success probability $P(y = 1|m, q)$ for all candidate models and route the query to the model with the highest probability.

Table 1: Model routing accuracy (%) comparison. **Micro** denotes the micro-averaged accuracy, calculated on the total correct predictions across all datasets. **Macro** denotes the macro-averaged accuracy, computed by first calculating the accuracy for each benchmark individually and then averaging these benchmark scores. We use **bold** to indicate the best results.

| Method | ASD | GPQA | GSM | Math | Logi | Med | Mmlu | Soci | PIQA | Tru | Micro | Macro |
|---|---|---|---|---|---|---|---|---|---|---|---|---|
| RouterDC | 62.1 | 21.2 | 70.5 | 40.5 | 41.2 | 54.5 | 70.9 | 30.9 | 80.6 | 50.0 | 54.9 | 52.2 |
| EmbedLLM | 34.9 | 24.8 | 82.1 | 47.7 | 31.4 | 65.8 | 80.1 | **37.0** | 86.6 | **52.7** | 60.2 | 54.3 |
| MODEL-SAT | 6.57 | 22.6 | 80.4 | 43.0 | 47.0 | 62.4 | 80.8 | 29.6 | 83.6 | 35.1 | 56.7 | 49.1 |
| Avengers-Pro | 12.1 | **29.4** | **89.3** | 55.3 | **49.0** | 72.9 | 83.2 | 30.9 | 86.6 | 41.9 | 62.1 | 55.7 |
| IrtNet (Ours) | **66.7** | 28.6 | **89.3** | **57.8** | **49.0** | **74.0** | **86.2** | 34.0 | **87.3** | 47.3 | **67.4** | **62.0** |

We compare our method with four advanced routing methods, RouterDC (Chen et al., 2024), EmbedLLM (Zhuang et al., 2025), MODEL-SAT (Zhang et al., 2025c) (with Qwen3-0.6B-Base (Yang et al., 2025) as base model), and Avengers-Pro (Zhang et al., 2025d). As shown in Table 1, our method achieves the best performance on both micro-average and macro-average metrics across 10 benchmarks. Specifically, IrtNet achieves a final micro-average accuracy of 67.4%, significantly outperforming strong baselines like EmbedLLM (60.2%) and Avengers-Pro (62.1%), which showcases its immense potential for the model routing task.

## 4.3 BENCHMARK PREDICTION

We use two settings to test the benchmark prediction ability of IrtNet: in-distribution (ID) and out-of-distribution (OOD). In the ID setting, while the two sets $\mathfrak{Q}$ and $Q$ have no overlap, they may still be closely related. In the OOD case, performance on a dataset may be predicted based on the model's performance on a different dataset.

**ID Correctness Prediction.** In this setting, we evaluate IrtNet's core capability for correctness prediction on model-query pairs. The results, presented in Table 2, highlight our model's exceptional data efficiency and predictive power.

Table 2: Correctness prediction accuracy (%) on different training data sizes.

| method | Dataset Size | | | | | | |
|---|---|---|---|---|---|---|---|
| | 1K | 5K | 10K | 15K | 20K | 25K | Full (29K) |
| KNN | 62.6 | 63.0 | 64.4 | 65.1 | 64.6 | 64.4 | 64.7 |
| EmbedLLM | 60.8 | 64.2 | 66.5 | 67.9 | 69.1 | 69.9 | 70.6 |
| IrtNet (ours) | **69.9** | **71.5** | **71.7** | **71.8** | **72.0** | **72.1** | **72.2** |

A key advantage of our framework is its ability to achieve strong performance with remarkably little training data. With just 1,000 queries—less than 4% of the full training set—IrtNet achieves a prediction accuracy of 69.9%. This performance significantly surpasses both the traditional KNN baseline (62.6%) and the state-of-the-art EmbedLLM (60.8%) trained on the same amount of data. Notably, our model's accuracy with only 1K queries already approaches the performance of EmbedLLM trained on the entire dataset (70.6%).

As the training set size increases, IrtNet consistently outperforms other baselines, reaching a final accuracy of 72.2% on the full training set. This demonstrates that our framework not only learns rapidly from limited samples but also scales effectively. The results strongly suggest that by modeling the interaction between model abilities and query characteristics, IrtNet captures their complex relationship more efficiently and accurately than existing methods.

**OOD Benchmark Prediction.** In this setting, we test IrtNet's ability to generalize its learned representations to make macroscopic performance predictions. The objective is to forecast an LLM's accuracy score on an entire benchmark it has never seen during training. This experiment uses a leave-one-out approach: we train IrtNet on all data except for one target benchmark and then use the trained model to predict the overall accuracy of all LLMs on that held-out benchmark. We exclude ASDiv and SocialIQA from this analysis, as the near-uniform scores across all models (approximately 0 and 0.3, respectively) suggest the data might represent noise rather than meaningful performance variation. This OOD setup serves as the ultimate test of whether our model representations can truly understand and generalize the abstract concept of LLM abilities.

Table 3: Root mean square error (RMSE) for OOD benchmark prediction. **Overall** denotes the total prediction error, calculated by treating all benchmarks as a single, unified test set.

| Method | GPQA | GSM | Math | Logi | Med | Mmlu | PIQA | Tru | Overall |
|---|---|---|---|---|---|---|---|---|---|
| EmbedLLM | 0.26 | 0.20 | 0.09 | 0.11 | 0.04 | 0.25 | 0.40 | **0.11** | 0.21 |
| IrtNet (Ours) | 0.26 | **0.19** | 0.09 | 0.11 | 0.04 | **0.19** | **0.36** | 0.12 | **0.19** |

The results in Table 3 demonstrate IrtNet's robust benchmark accuracy prediction. Across the eight benchmarks, IrtNet achieves a lower root mean square error (RMSE) than EmbedLLM on three datasets and performs equally on four. This leads to an overall RMSE of 0.19, representing a nearly 10% error reduction over EmbedLLM's 0.21.

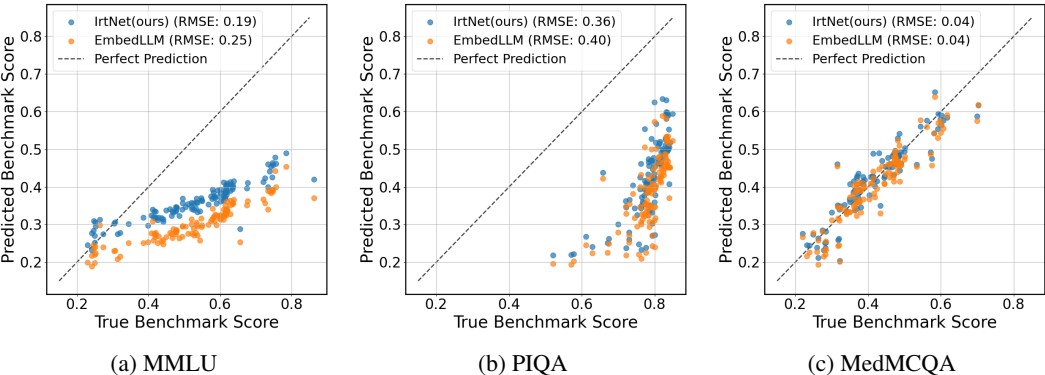

| (a) MMLU | (b) PIQA | (c) MedMCQA |
|---|---|---|

Figure 3: Predicted vs. true benchmark scores (in [0-1]) on three OOD benchmarks. The scatter plots represent the predicted LLM scores by IrtNet and EmbedLLM. IrtNet's predictions (blue dots) align more closely with the perfect prediction diagonal line on MMLU and PIQA, which means lower prediction errors. IrtNet and EmbedLLM are tied on MedMCQA with the predicted scores almost coinciding with the true scores.

Figure 3 presents the specific prediction distributions on three representative benchmarks (MMLU, PIQA, and MedMCQA), vividly illustrating how IrtNet's predictions achieve a smaller prediction error compared to EmbedLLM or cluster very tightly around the true scores.

## 4.4 ABLATION STUDY (REVISED)

In this section, we investigate the necessity of both the MoE architecture and the IRT formulation. We ablate the MoE layer by replacing it with an MLP network of the same number of parameters and ablate the IRT formulation by replacing it with logistic regression.

As shown in Table 4, after removing the MoE layer or IRT formulation, IrtNet exhibits a significant performance drop in the model routing task and a slight decline in the correctness prediction task.

Table 4: Ablation study on the MoE layer and IRT formulation of IrtNet. **Routing** denotes the model routing task and **Correctness** denotes the correctness prediction task.

| Task | IrtNet | w/o MoE | w/o IRT | w/o Both (EmbedLLM) |
|---|---|---|---|---|
| **Routing** | 67.4 | $64.0_{\downarrow 3.4}$ | $63.8_{\downarrow 3.6}$ | $60.2_{\downarrow 6.2}$ |
| **Correctness** | 72.2 | $71.3_{\downarrow 0.9}$ | $71.1_{\downarrow 1.1}$ | $70.6_{\downarrow 1.6}$ |

This ablation result fully demonstrates the effectiveness of the MoE structure and the IRT formulation. Furthermore, the more substantial decline in the model routing task indicates that IrtNet is particularly effective at understanding the relative ranking among models. This is likely because IrtNet produces a more discriminative $\alpha_q$ representation, which more accurately captures the distinctions among model abilities.

## 5 INTERPRETABILITY ANALYSIS

### 5.1 UNDERSTANDING DISCRIMINATION

In our framework, the discrimination vector $\alpha_q$ provides a rich, quantitative profile of a query's intrinsic characteristic. A high value in a particular dimension of $\alpha_q$ signifies that the query places a strong demand on the corresponding latent ability, making a model's proficiency in that dimension more critical for a correct answer. Thus, the discrimination vector reflects which abilities a query is designed to test and how strongly it tests them. To validate this interpretation, we conduct a visualization experiment on the test set. We use the trained IrtNet to compute the discrimination vectors for queries on the test set and then project these high-dimensional vectors into a two-dimensional space using t-SNE.

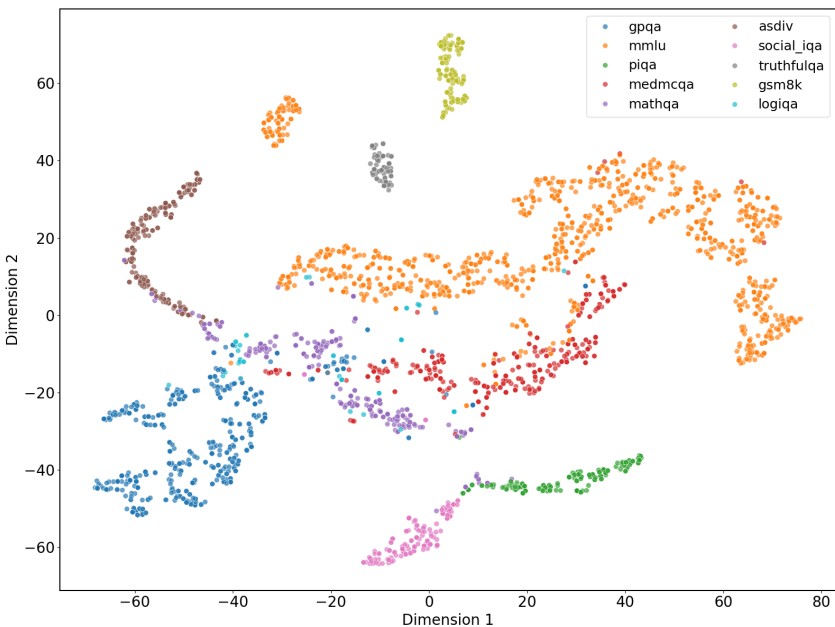

Figure 4: T-SNE visualization of learned query discrimination vectors $\alpha_q$.

Remarkably, despite IrtNet never being exposed to any dataset labels during training, the resulting visualization in Figure 4 reveals a clear and well-defined clustering structure. We observe that queries originating from the same benchmark naturally form distinct semantic groups in this learned space. This provides strong evidence that the learned discrimination vector $\alpha_q$ has successfully captured the unique demands of different query types. The spontaneous emergence of these clusters demonstrates that IrtNet has effectively modeled the distinct discriminative properties inherent to each semantic query group, mapping them into an interpretable space.

## 5.2 VALIDATING DIFFICULTY

We perform a quantitative analysis to validate that our learned difficulty parameter $\beta_q$ is a meaningful measure of a query's intrinsic challenge. For our experimental setup, we first establish an objective difficulty standard for each benchmark. This standard is defined as the average ground-truth accuracy achieved by the entire pool of 112 models on that dataset. A lower average score naturally corresponds to a higher objective difficulty. We then compare this objective standard against the average learned difficulty parameter $\beta_q$ for each respective benchmark.

Table 5: Comparison of average model accuracy (in [0, 1]) and learned difficulty $\beta_q$ across benchmarks.

|  | ASD | GPQA | Logi | Math | Soci | Tru | Med | GSM | Mmlu | PIQA |
|---|---|---|---|---|---|---|---|---|---|---|
| Average Accuracy | 0.04 | 0.21 | 0.29 | 0.33 | 0.34 | 0.36 | 0.42 | 0.42 | 0.53 | 0.78 |
| Average $\beta_q$ | 1.69 | 0.71 | 0.45 | 0.40 | 0.33 | 0.32 | 0.16 | 0.17 | -0.08 | -0.78 |

As shown in Table 5, our findings reveal an exceptionally strong negative correlation between the two: as the average model accuracy on a benchmark decreases (i.e., it gets harder), the learned $\beta_q$ value consistently increases. Quantitatively, the Pearson correlation coefficient between the objective average accuracy and the learned average difficulty $\beta_q$ is -0.9721. This near-perfect correlation provides compelling evidence that the difficulty parameter learned by IrtNet is not an arbitrary value, but a valid and reliable metric that accurately captures the empirical hardness of the queries.

## 5.3 PROBING LLM EMBEDDING

To investigate whether the learned LLM embedding $\theta_m$ captures meaningful model characteristics, we conduct a similarity validation experiment. Our hypothesis is that if the vectors are coherent, models sharing fundamental traits should be geometrically closer in the learned space. To test this, we partition our pool of models into distinct groups based on two criteria: model family, such as the Qwen and Llama families, and domain specialization for tasks like medicine, code, and math. We then calculate and compare two metrics: the average intra-community L2 distance, which measures the distance between models within a single group, and the average inter-community L2 distance, which measures the distance from that group's models to all outside models.

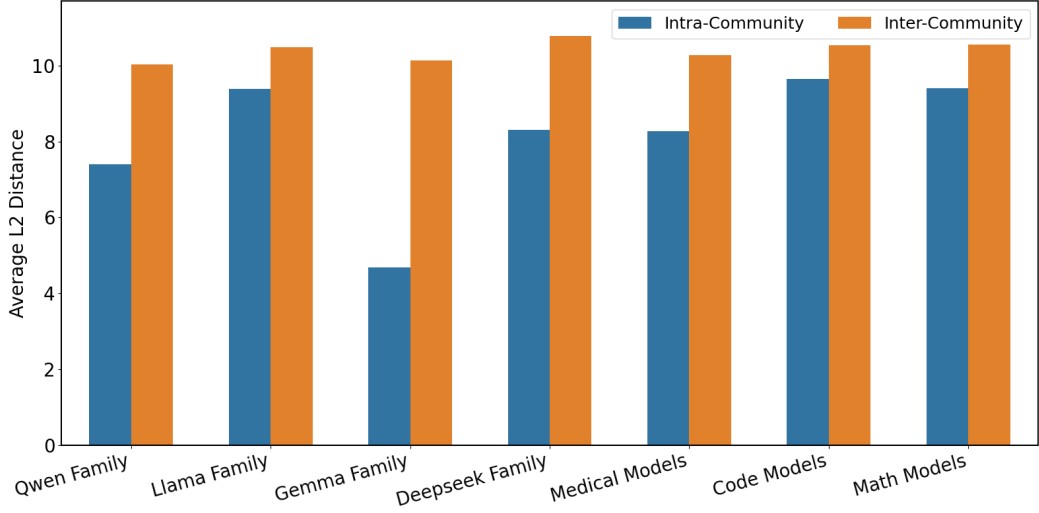

Figure 5: Comparison of intra-community and inter-community L2 distances for LLM embeddings.

Figure 5 provides compelling evidence for our hypothesis. Across all defined groups, the average intra-community distance is consistently and significantly smaller than the average inter-community distance. This clear geometric clustering holds true for models grouped by both architectural lineage, such as the Llama and Gemma families, or by functional specialization, like the groups of Code and

Math models. This finding demonstrates that model embedding $\theta_m$ is a meaningful, well-structured representation that effectively encodes a model's specialized abilities.

# 6 RELATED WORK

**Representation Learning**  The concept of learning compact vector representations for complex, high-dimensional objects is a well-established paradigm in machine learning . For instance, models like SentenceTransformer (Reimers & Gurevych, 2019) and Qwen3-Embedding (Zhang et al., 2025b) effectively embed sentences into a dense embedding space; in knowledge graphs, methods like TransE (Bordes et al., 2013) learn low-dimensional embeddings for entities and relations. More recently, EmbedLLM (Zhuang et al., 2025) generalizes this paradigm to model LLM abilities with an encoder-decoder approach, demonstrating that learning a compact representation of LLM abilities can facilitate multiple downstream tasks. Our work is most closely related to EmbedLLM, but bears key conceptual and methodological differences. We apply IRT to model LLM abilities, providing a theory-driven approach to this paradigm, and introduce an MoE-based method to explicitly learn query characteristics (discrimination and difficulty) that EmbedLLM does not capture. Furthermore, our method leads to significant outperformance compared to EmbedLLM.

**Model Routing**  Model routing for LLMs (Shnitzer et al., 2023; Lu et al., 2024; Srivatsa et al., 2024) has emerged as a critical strategy for efficiently managing a diverse suite of models. Recent research in this field generally falls into two streams. The first emphasizes striking a balance between performance and computational efficiency (Jiang et al., 2023; Ong et al., 2024; Feng et al., 2024; Wang et al., 2025; Zhang et al., 2025d). The second prioritizes pushing model performance to the highest possible levels (Lu et al., 2024; Chen et al., 2024; Zhang et al., 2025c;e). RouterDC (Chen et al., 2024) proposes a dual-contrastive learning framework to better align queries and model representations. Model-SAT (Zhang et al., 2025c) creates a capability representation for each model through a lightweight aptitude test to select the most suitable model. The Avengers (Zhang et al., 2025e) presents a training-free, clustering-based routing framework that selects the optimal model. While our method does not specifically target model routing, the router developed using our approach significantly outperforms state-of-the-art routing methods.

**Benchmark Prediction**  Evaluating LLMs on large benchmarks requires substantial computational and financial cost, which has motivated research on benchmark prediction that seeks to estimate overall performance from only a subset of representative data. Core-set selection methods such as Anchor Points (Vivek et al., 2023) attempt to identify a small number of informative queries that preserve model rankings nearly as well as full benchmarks. Other work has shown that even simple random subsets combined with regression models can provide surprisingly strong estimates of average accuracy (Zhang et al., 2025a). More recently, IRT-based methods such as tinyBenchmarks (Polo et al., 2024b) have demonstrated that psychometric modeling can reduce evaluation costs even further. While our approach to learning representations of model abilities can be applied to benchmark prediction, it differs from the above studies that focus on data selection strategies. Distinct from these, observational scaling laws (Polo et al., 2024a; Ruan et al., 2024) predict aggregate benchmark performance by modeling latent skills from model metadata. Unlike these macro-level approaches, we leverage query-level semantic prediction and then aggregate the results across the entire benchmark.

# 7 CONCLUSION

In this paper, we pioneer the application of item response theory (IRT) to formally model LLM abilities. We introduce IrtNet, a IRT-based framework that learns compact representations of LLM abilities. Based on a Mixture-of-Experts architecture, it jointly learns model embeddings alongside query difficulty and discrimination. Extensive Experiments demonstrate that IrtNet sets a new state-of-the-art in model routing and achieves highly data-efficient, more accurate benchmark prediction. Furthermore, the learned representations are also interpretable. The difficulty parameter strongly correlates with empirical results, and that model embeddings naturally cluster by family and function. Overall, IrtNet provides a robust and insightful tool for effective model evaluation, selection, and analysis in the growing LLM ecosystem.

## REPRODUCIBILITY STATEMENT

To facilitate the reproduction of our results, we have provided the implementation of our method in the supplementary materials. The datasets used in this study are all publicly accessible, and details can be found in the experiments section. Furthermore, to contribute to the open-source community, we pledge to release a cleaned, user-friendly, and well-documented version of our source code on a public repository (e.g., GitHub) upon acceptance of the manuscript.

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

# A  MORE EXPERIMENTAL RESULTS (ALL NEW)

## A.1  DETAILED EXPERIMENTAL SETTING AND COMPUTATIONAL COST

We trained our IrtNet for 15 epochs with a learning rate of $1e-4$, a weight decay of $1e-4$, a batch size of 2048, and a dropout rate of 0.5.

We used a single NVIDIA A100 (80GB) GPU. Training with a batch size of 2048 consumed approximately 2GB of VRAM and took a total of 15 minutes for 15 epochs.For inference, when the batch size was set equal to the number of models (112), the VRAM consumption was 1GB. Processing 3000 test samples took 20 seconds, resulting in an average time of 6.7ms per test sample.

## A.2  MODEL ROUTING

### A.2.1  MORE DATASETS

To verify the applicability of IrtNet on more datasets, we introduce 10 datasets used by Avengers (Zhang et al., 2025e), including BBH (Suzgun et al., 2023), EmoryNLP (Byrkjeland et al., 2018), FinQA (Chen et al., 2021), HumanEval (Wang et al., 2024), K&K (Xie et al., 2024), Livecodebench (Jain et al., 2024), Math500 (Lightman et al., 2023), MBPP (Austin et al., 2021), MELD (Poria et al., 2019), and MMLUPro (Wang et al., 2024), totaling 8,550 questions. Of these, 6,840 questions (80%) were used for the training set and 1,710 questions (20%) for the test set.

In this supplementary experiment, we use more recent LLMs compared to those in the main text, specifically 20 models with 7-9B parameters. These models are: DeepHermes-3-Llama-3-8B-Preview, DeepSeek-R1-0528-Qwen3-8B, DeepSeek-R1-Distill-Qwen-7B, Fin-R1, GLM-Z1-9B-0414, Intern-S1-mini, Llama-3.1-8B-Instruct, Llama-3.1-8B-UltraMedical, Llama-3.1-Nemotron-Nano-8B-v1, MiMo-7B-RL-0530, MiniCPM4.1-8B, NVIDIA-Nemotron-Nano-9B-v2, OpenThinker3-7B, Qwen2.5-Coder-7B-Instruct, Qwen3-8B, cogito-v1-preview-llama-8B, gemma-2-9b-it, glm-4-9b-chat, granite-3.3-8b-instruct, and internlm3-8b-instruct.

Table 6: Model routing accuracy (%) comparison on Avengers datasets. **Micro** denotes the micro-averaged accuracy, calculated on the total correct predictions across all datasets. **Macro** denotes the macro-averaged accuracy, computed by first calculating the accuracy for each benchmark individually and then averaging these benchmark scores. We use **bold** to indicate the best results.

| Method | Bbh | Emory | Finqa | HE | K&K | Lcb | M500 | Mbpp | Meld | MP | Micro | Macro |
|--------|------|------|------|------|------|------|------|------|------|------|------|------|
| RouterDC | 65.4 | **46.4** | 59.1 | 68.0 | 80.6 | 16.7 | 91.3 | 63.5 | 54.6 | 55.7 | 56.9 | 60.1 |
| ModelSAT | **90.1** | 41.2 | **73.7** | 72.0 | 73.9 | **72.2** | 93.3 | 74.7 | 52.2 | 70.7 | 70.7 | 71.4 |
| EmbedLLM | 87.4 | 44.0 | 70.5 | 70.0 | 70.6 | 67.8 | **95.3** | 70.0 | 52.2 | **71.7** | 69.0 | 69.9 |
| Avengers-Pro | 88.9 | 42.6 | **73.7** | 72.0 | 75.4 | 69.7 | 94.7 | 75.4 | **56.5** | 69.7 | 71.2 | 71.8 |
| IrtNet (Ours) | 89.8 | 44.5 | 73.4 | **78.0** | **81.0** | 69.7 | 94.0 | **75.8** | 55.7 | **71.7** | **72.1** | **73.4** |

As shown in Table 6, our method achieves the best performance on both micro-average and macro-average accuracy across 10 benchmarks. Combining the experiments in Table 1, IrtNet achieves SOTA routing results on a total of 20 datasets, fully demonstrating the versatility of our method.

### A.2.2  NOISY SETTING

To investigate the effect of noisy data on IrtNet, we adopt the original dataset from EmbedLLM. In this original dataset (including both the training and test sets), the same question might be simultaneously assigned multiple labels of both 0 and 1. The experimental results, shown in Table 7, indicate that under noisy data, when compared to the results in Table 1, our routing performance does not decline and still maintain the SOTA level, demonstrating that IrtNet can function effectively even in noisy environments.

Table 7: Model routing accuracy (%) comparison on Embedllm dataset with noisy labels.

| Method | ASD | GPQA | GSM | Math | Logi | Med | Mmlu | Soci | PIQA | Tru | Micro | Macro |
|---|---|---|---|---|---|---|---|---|---|---|---|---|
| RouterDC | 64.7 | 26.6 | 70.5 | 38.4 | 39.2 | 59.3 | 72.9 | 27.8 | 81.3 | 46.0 | 56.9 | 52.7 |
| EmbedLLM | 48.0 | 28 | 82.1 | 48.5 | 43.1 | 65.5 | 80.8 | **35.2** | **88.1** | 50.0 | 62.0 | 56.9 |
| MODEL-SAT | 8.1 | 11.0 | 67.9 | 39.7 | 45.1 | 56.5 | **86.3** | 32.1 | 86.6 | 33.8 | 55.8 | 46.7 |
| Avengers-Pro | 13.6 | 12.8 | **89.3** | 55.7 | **49.0** | 72.3 | 84.0 | 30.9 | 86.6 | 41.9 | 59.7 | 53.6 |
| IrtNet (Ours) | **66.7** | **27.8** | **89.3** | **57.0** | 47.1 | **75.4** | 86.2 | 30.9 | 85.8 | **52.7** | **67.5** | **62.7** |

## A.3 BENCHMARK PREDICTION

### A.3.1 MORE BASELINES

In this section, we consider comparing additional benchmark prediction methods. We primarily evaluate three approaches: Sloth (Polo et al., 2024a), observational scaling laws (Ruan et al., 2024), and Random-Sampling-Learn (Zhang et al., 2025a). We focus on comparing the latter two because Sloth requires two key inputs: model parameters and the number of training tokens. While parameter counts are accessible for most open-sourced models, the specific number of training tokens is often unavailable.

We maintain the OOD experimental setup to evaluate the RMSE between the predicted and actual benchmark scores across all methods. To enable the implementation of OSL and RSL, we have to reserve an additional 5% of the models, utilizing their performance on 50 cold-start queries along with their ground-truth full-benchmark scores for training. In contrast, EmbedLLM and IrtNet do not require this additional supervision, as they rely solely on query semantics and model identifiers for prediction.

Table 8: Root mean square error (RMSE) for OOD benchmark prediction. **Avg.** denotes the average RMSE of all benchmarks.

| Method | GPQA | GSM | Math | Logi | Med | Mmlu | PIQA | Tru | Avg. |
|---|---|---|---|---|---|---|---|---|---|
| OSL (Ruan et al., 2024) | 0.30 | 0.36 | **0.06** | 0.21 | 0.28 | 0.45 | 0.51 | 0.36 | 0.32 |
| RSL (Zhang et al., 2025a) | **0.23** | 0.43 | 0.28 | **0.08** | 0.27 | 0.37 | **0.35** | 0.29 | 0.29 |
| EmbedLLM (Zhuang et al., 2025) | 0.26 | 0.20 | 0.09 | 0.11 | 0.04 | 0.25 | 0.40 | **0.11** | 0.18 |
| IrtNet (Ours) | 0.26 | **0.19** | 0.09 | 0.11 | 0.04 | **0.19** | 0.36 | 0.12 | **0.16** |

The results are presented in Table 8. It is evident that OSL and RSL exhibit significantly higher prediction errors compared to EmbedLLM and IrtNet. This disparity primarily stems from the difference in prediction granularity: OSL and RSL perform benchmark-level predictions by inferring the relationship between few-shot observations and full-benchmark scores based on historical models, whereas EmbedLLM and IrtNet achieve significantly greater precision through fine-grained, query-level prediction.

### A.3.2 MORE RESULTS IN TABLE 3

In this subsection, we additionally present the result figures for the remaining five OOD Benchmark predictions that were not shown in Section 4.3. As shown in Figure 6, on these five datasets, the prediction results of IrtNet and EmbedLLM are essentially tied, and they exhibit the same bias, meaning they either simultaneously overestimate or underestimate the results for a specific dataset.

## A.4 ABLATION STUDY

### A.4.1 QUERY ENCODER

We perform an ablation study on the query encoder, replacing all-mpnet-base-v2 with the relatively weaker bert-base-nli-mean-tokens (Reimers & Gurevych, 2019). As shown in Table 9, compared to the results in Table 1, all methods experience performance fluctuations. Except for ModelSAT, which sees a performance increase, all other methods show a decline in performance. This suggests

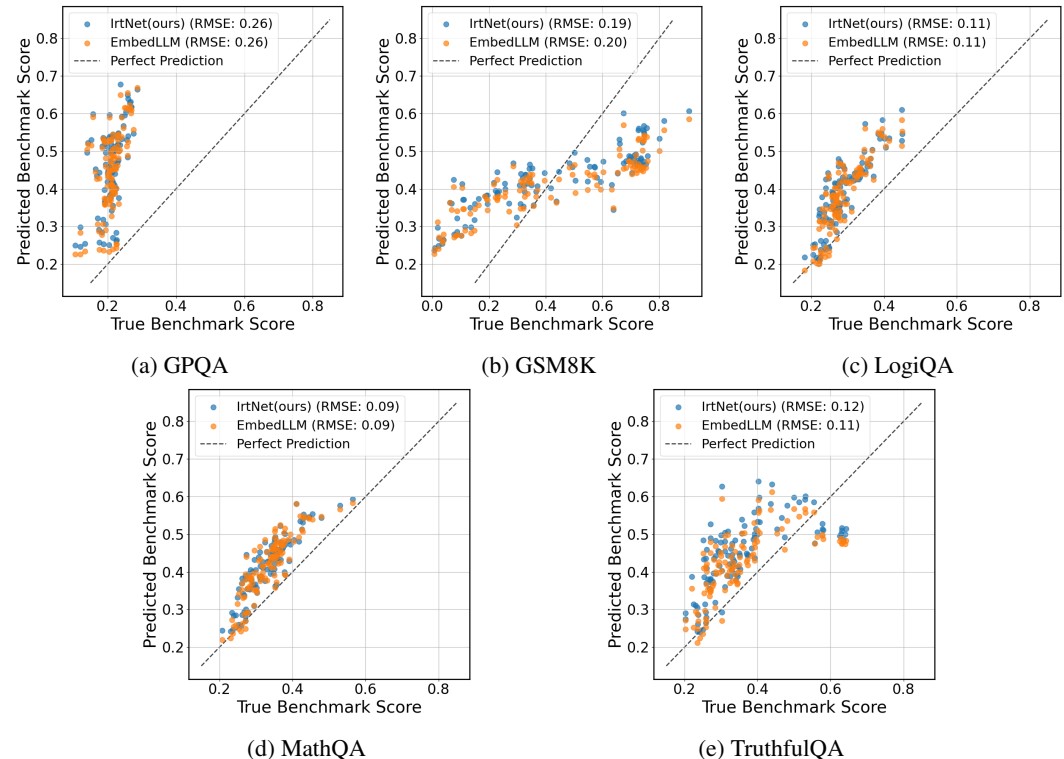

Figure 6: Predicted vs. true benchmark scores (in [0-1]) on five OOD benchmarks. The scatter plots represent the predicted LLM scores by IrtNet and EmbedLLM.

that routing methods are indeed dependent on the effectiveness of the query encoder to some extent. Overall, we still maintain strong competitiveness, achieving first place in Micro Accuracy and second in Macro Accuracy.

Table 9: Model routing accuracy (%) comparison with bert-base-nli-mean-tokens as query encoder.

| Method | ASD | GPQA | GSM | Math | Logi | Med | Mmlu | Soci | PIQA | Tru | Micro | Macro |
|--------|-----|------|-----|------|------|-----|------|------|------|-----|-------|-------|
| RouterDC | 36.9 | 16.6 | 21.4 | 40.9 | 39.2 | 59.0 | 65.8 | 28.4 | 81.3 | 46.0 | 49.0 | 43.6 |
| EmbedLLM | 15.7 | 22.0 | **88.4** | 54.4 | 47.1 | 67.5 | 79.1 | 30.3 | 85.8 | 48.7 | 58.8 | 53.9 |
| MODEL-SAT | **70.2** | 26.6 | 86.6 | 49.0 | 45.1 | 69.5 | 82.3 | 29.6 | 85.1 | **50.0** | 64.1 | **59.4** |
| Avengers-Pro | 11.1 | **28.6** | 88.4 | 54.4 | **49.0** | **73.2** | 81.7 | **30.9** | **86.6** | 35.1 | 61.0 | 53.9 |
| IrtNet (Ours) | 61.6 | 25.8 | 86.6 | 53.2 | 41.2 | 70.6 | **85.0** | 29.6 | 85.8 | **50.0** | **64.9** | 58.9 |

### A.4.2 MODEL EMBEDDING DIMENSION

In this section, we investigate the sensitivity of IrtNet to the dimension of the LLM's compact representation. We only vary the dimension of the learned model embedding and test the performance on both the model routing and benchmark prediction tasks.

Table 10: Sensitivity of IrtNet to the dimension of the LLM Embedding. **Routing** denotes the model routing task and **Correctness** denotes the correctness prediction task.

| Task | LLM Embedding Dimension | | | | | | |
|------|------|------|------|------------|------|------|------|
| | 32 | 64 | 128 | 232 (Ours) | 512 | 1024 | 2048 |
| **Routing** | 64.8 | 65.9 | 66.9 | 67.4 | 66.7 | 67.2 | **67.6** |
| **Correctness** | 71.8 | 71.9 | 72.0 | **72.2** | 72.1 | 72.0 | 72.0 |

As shown in Table 10, using 232 dimensions as the baseline (chosen for consistency with Em-bedLLM), when we decrease the embedding dimension, the performance of both tasks significantly drop. This is primarily because an overly small dimension cannot capture sufficient information. Conversely, when we continue to increase the model embedding dimension, the performance barely improves. This suggests that due to limited training data, further increasing the dimension becomes redundant once a suitable dimension is sufficient to fully represent the information.

### A.4.3   MoE Expert Number

In this subsection, we investigate the effect of MoE sparsity/density and the influence of the number of experts. We adopted two settings: In the dense setting, we simultaneously varied the total number of experts and the number of activated experts. In the sparse setting, we kept the total number of experts at 40 and then varied the number of activated experts.

As shown in Table 11, due to our limited data size, the sparse MoE indeed suffered from insufficient training, and the model's performance gradually improved as the number of activated experts increased. In the dense setting, the model's performance change was largely insignificant.

Table 11: Ablation study of MoE expert number. We use **bold** to indicate the best results.

| MoE Architecture | Total Experts | Active Experts | Routing | Correctness |
|---|---|---|---|---|
| Dense (Ours) | 40 | 40 | **67.4** | **72.2** |
| Sparse | 40 | 4 | $62.4_{\downarrow 5.0}$ | $68.7_{\downarrow 3.5}$ |
| Sparse | 40 | 20 | $66.1_{\downarrow 1.3}$ | $71.2_{\downarrow 1.0}$ |
| Dense | 20 | 20 | $67.0_{\downarrow 0.4}$ | $72.0_{\downarrow 0.2}$ |
| Dense | 80 | 80 | $66.7_{\downarrow 0.7}$ | $72.0_{\downarrow 0.2}$ |

## A.5   Interpretability Analysis

### A.5.1   Understanding Discrimination

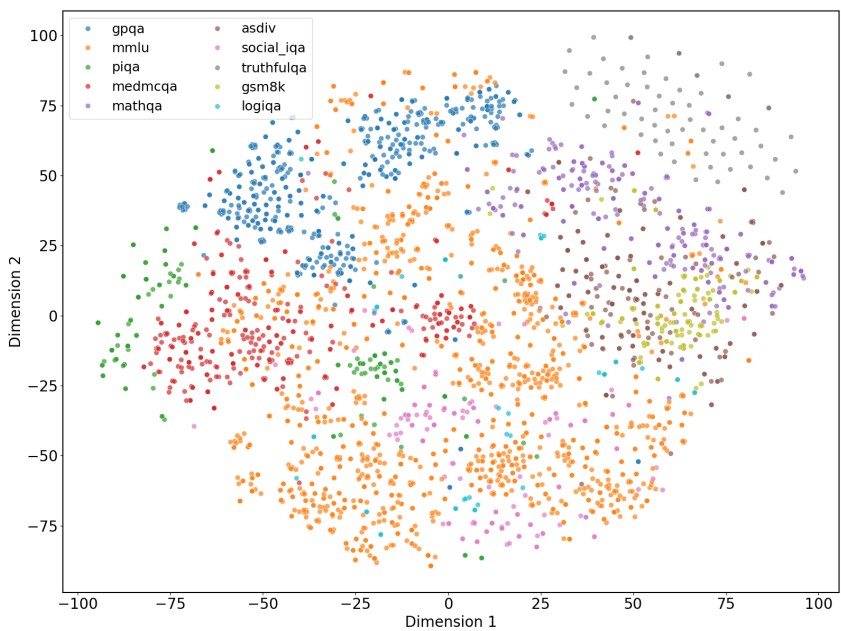

Figure 7: T-SNE visualization of raw query embeddings.

As shown in Figure 7, we perform the t-SNE visualization on the raw Sentence-BERT embeddings ($\mathbf{v}_q$). While the raw $\mathbf{v}_q$ embeddings show a general clustering tendency, the clusters are often diffuse and overlapping. In contrast, the learned discrimination vectors $\alpha_q$ in Figure 4 exhibit significantly

tighter and clearer separation between different benchmark groups. This comparison confirms that our MoE-based network successfully acts as a structural filter, refining the generic semantic features of $\mathbf{v}_q$ into highly distinguishable and task-specific latent requirements. This refinement is the core reason for IrtNet's superior routing and benchmark prediction performance.

### A.5.2 VALIDATING DIFFICULTY

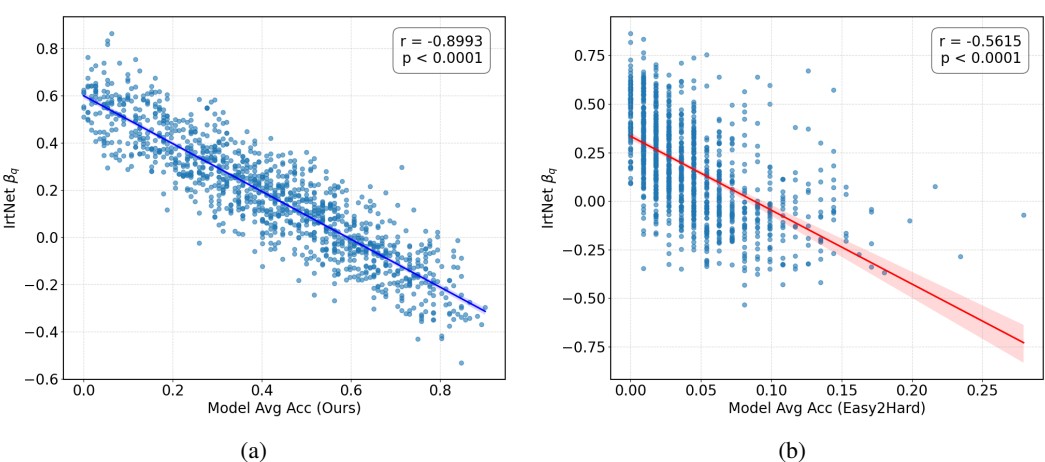

(a)                                                                      (b)

Figure 8: The correlation between the average model accuracy per question on the GSM8K dataset and the difficulty parameter $\beta_q$. (a) shows the correlation between the accuracy of the models used in our experiment and $\beta_q$, and (b) shows the correlation between the Easy2Hard-Bench model accuracy and $\beta_q$.

To further demonstrate the significance of the difficulty parameter learned by IrtNet, we align the GSM8K results in our experiments with the Easy2Hard-Bench (Ding et al., 2024). We calculate the correlation between the average accuracy of our model set and the learned difficulty parameter $\beta_q$, as well as the correlation between the average accuracy of the Easy2Hard-Bench model set on GSM8K and $\beta_q$.

As shown in Figure 8, the average accuracy of our model set exhibits an extremely strong negative correlation with the difficulty parameter $\beta_q$, with a correlation coefficient $r = -0.8993$. The average accuracy per question in the Easy2Hard-Bench model set also shows a moderate negative correlation with $\beta_q$, with a correlation coefficient $r = -0.5615$. We believe it is reasonable for the Easy2Hard-Bench correlation to be lower than the correlation within our own dataset because $\beta_q$ fundamentally learns the average performance of the models in the training data, and the average performance of the Easy2Hard-Bench model set differs from that of our model set. This explains why our $\beta_q$ correlates exceptionally well with the average performance of the models in our own dataset.

Concurrently, we also categorize the questions into five difficulty levels—Easiest, Easy, Medium, Hard, and Hardest—based on the model accuracy in Easy2Hard-Bench, and sampled representative questions for layered display. As shown in Table 12, the accuracies for the "Easy" and "Hard" questions are even reversed in our dataset: they are 0.063 and 0.018 in Easy2Hard-Bench, respectively, but 0.32 and 0.38 in ours. This is why the correlation between $\beta_q$ and Easy2Hard-Bench accuracy is lower. In contrast, $\beta_q$ correlates very well negatively with our own accuracy: the higher the accuracy, the lower the difficulty $\beta_q$.

### A.6 MODEL LIST

Here is an exhaustive list of models contained in Section 4:

Table 13: The exhaustive list of the 112 models used in Section 4.

| Model Name | Model Name |
| --- | --- |
| meta-llama/LlamaGuard-7b | meta-llama/Llama-2-13b-chat-hf |

Table 13: The exhaustive list of the 112 models used in Section 4.

| Model Name | Model Name |
| --- | --- |
| 01-ai/Yi-34B-Chat | meta-llama/Llama-2-70b-chat-hf |
| WizardLM/WizardLM-70B-V1.0 | allenai/tulu-2-dpo-70b |
| lmsys/vicuna-13b-v1.5 | lmsys/vicuna-33b-v1.3 |
| Qwen/Qwen-14B-Chat | upstage/SOLAR-10.7B-Instruct-v1.0 |
| openchat/openchat-3.5-0106 | openchat/openchat-3.5 |
| berkeley-nest/Starling-LM-7B-alpha | HuggingFaceH4/zephyr-7b-beta |
| TheBloke/tulu-30B-fp16 | mistralai/Mistral-7B-Instruct-v0.1 |
| tiiuae/falcon-40b-instruct | lmsys/vicuna-13b-v1.5-16k |
| codellama/CodeLlama-34b-Instruct-hf | TheBloke/WizardLM-13B-V1.2-GGUF |
| lmsys/vicuna-7b-v1.5 | NousResearch/Nous-Hermes-13b |
| project-baize/baize-v2-13b | lmsys/vicuna-7b-v1.5-16k |
| mosaicml/mpt-30b-instruct | meta-llama/Llama-2-7b-chat-hf |
| TheBloke/koala-13B-HF | nomic-ai/gpt4all-13b-snoozy |
| h2oai/h2ogpt-gm-oasst1-en-2048-open-llama-13b | mosaicml/mpt-7b-chat |
| databricks/dolly-v2-12b | stabilityai/stablelm-tuned-alpha-7b |
| OpenAssistant/oasst-sft-4-pythia-12b-epoch-3.5 | deepseek-ai/deepseek-llm-67b-chat |
| NousResearch/Nous-Hermes-2-Yi-34B | CausalLM/34b-beta |
| SUSTech/SUS-Chat-34B | SUSTech/SUS-Chat-72B |
| Qwen/Qwen-72B | Intel/neural-chat-7b-v3-3 |
| ibivibiv/alpaca-dragon-72b-v1 | JaeyeonKang/CCK-Asura-v1 |
| ConvexAI/Luminex-34B-v0.2 | ConvexAI/Luminex-34B-v0.1 |
| CorticalStack/pastiche-crown-clown-7b-dare-dpo | ogno-monarch-jaskier-merge-7b-OH-PREF-DPO |
| bardsai/jaskier-7b-dpo-v5.6 | FelixChao/Scorpio-7B |
| dfurman/HermesBagel-34B-v0.1 | kevin009/llamaRAGdrama |
| sail/Sailor-7B | AiMavenAi/Prometheus-1.3 |
| Q-bert/Optimus-7B | cognitivecomputations/yayi2-30b-llama |
| zhengr/MixTAO-7Bx2-MoE-v8.1 | fblgit/UNA-SimpleSmaug-34b-v1beta |
| mistralai/Mixtral-8x7B-Instruct-v0.1 | microsoft/Orca-2-13b |
| EleutherAI/pythia-12b | cloudyu/Mixtral-11Bx2-MoE-19B |
| rishiraj/CatPPT-base | Deci/DeciLM-7B |
| microsoft/phi-2 | scb10x/typhoon-7b |
| 01-ai/Yi-6B-200K | 01-ai/Yi-6B |
| TigerResearch/tigerbot-13b-base | augmxnt/shisa-base-7b-v1 |
| microsoft/phi-1.5 | golaxy/gowizardlm |
| bigscience/bloom-7b1 | mlabonne/AlphaMonarch-7B |
| CultriX/NeuralTrix-bf16 | shadowml/MBeagleX-7B |
| yam-peleg/Experiment26-7B | deepseek-ai/deepseek-math-7b-instruct |
| meta-math/MetaMath-Mistral-7B | kyujinpy/Sakura-SOLRCA-Math-Instruct-DPO-v1 |
| FelixChao/llama2-13b-math1.2 | Plaban81/Moe-4x7b-math-reason-code |
| MaziyarPanahi/WizardLM-Math-70B-v0.1 | abhishek/zephyr-beta-math |
| meta-math/MetaMath-Llemma-7B | EleutherAI/llemma-34b |
| EleutherAI/llemma-7b | FelixChao/vicuna-7B-physics |
| Harshvir/Llama-2-7B-physics | FelixChao/vicuna-7B-chemical |
| BioMistral/BioMistral-7B | BioMistral/BioMistral-7B-DARE |
| PharMolix/BioMedGPT-LM-7B | Biomimicry-AI/ANIMA-Nectar-v2 |
| codellama/CodeLlama-7b-hf | codellama/CodeLlama-13b-Instruct-hf |
| deepseek-ai/deepseek-coder-1.3b-base | deepseek-ai/deepseek-coder-6.7b-instruct |
| OpenBuddy/openbuddy-codellama2-34b-v11.1-bf16 | TheBloke/CodeLlama-70B-Instruct-AWQ |
| AdaptLLM/medicine-chat | AdaptLLM/medicine-LLM |
| AdaptLLM/medicine-LLM-13B | Writer/palmyra-med-20b |
| SciPhi/SciPhi-Self-RAG-Mistral-7B-32k | Neko-Institute-of-Science/metharme-7b |
| Neko-Institute-of-Science/pygmalion-7b | SciPhi/SciPhi-Mistral-7B-32k |
| shleeeee/mistral-ko-tech-science-v1 | codefuse-ai/CodeFuse-DeepSeek-33B |
| WizardLM/WizardCoder-Python-34B-V1.0 | bigcode/octocoder |
| meta-llama/Meta-Llama-3-8B | meta-llama/Meta-Llama-3-8B-Instruct |
| meta-llama/Meta-Llama-3-70B | meta-llama/Meta-Llama-3-70B-Instruct |

Table 13: The exhaustive list of the 112 models used in Section 4.

| Model Name | Model Name |
|---|---|
| meta-llama/Meta-Llama-Guard-2-8B | Qwen/Qwen1.5-32B-Chat |
| Qwen/Qwen1.5-4B-Chat | Qwen/Qwen1.5-0.5B-Chat |
| Qwen/Qwen1.5-7B-Chat | Nexusflow/Starling-LM-7B-beta |
| google/gemma-7b-it | google/gemma-2b-it |

Table 12: Case study of question accuracy and difficulty parameter $\beta_q$.

| E2H Difficulty | Question | E2H Acc | Our Acc | IrtNet $\beta_q$ |
|---|---|---|---|---|
| Easiest | A raspberry bush has 6 clusters of 20 fruit each and 67 individual fruit scattered across the bush. How many raspberries are there total? | 0.2793 | 0.7054 | -0.0716 |
| Easy | It's strawberry-picking time on Grandma Concetta's farm. Tony can pick 6 quarts of strawberries per hour, while Bobby picks one less quart of strawberries per hour than Tony. Kathy can pick twice as many strawberries per hour as Bobby, and Ricky picks two fewer quarts of strawberries per hour than does Kathy. In total, how many quarts of strawberries can Tony, Bobby, Ricky, and Kathy pick per hour on Grandma Concetta's farm? | 0.0631 | 0.3214 | 0.3380 |
| Medium | Laurel's friend gave her 24 baby outfits that her child no longer needed. At her baby shower, Laurel received twice the amount of new baby outfits. Then, Laurel's mom gifted her with another 15 baby outfits. How many outfits does she have for her baby? | 0.0360 | 0.4643 | 0.1699 |
| Hard | Oscar has 24 lollipops and eats 2 on his way to school. He passes 14 out to his friends. He buys twice as many lollipops on his way home as he gave to his friends. He eats 3 more that night and 2 more in the morning. How many lollipops does Oscar have? | 0.0180 | 0.3839 | 0.2611 |
| Hardest | For a New Year's resolution, Andy wants to lose 30 lbs. by his birthday, which is July 19th. Today is December 31st. If Andy needs to burn 3500 calories to lose a pound, how much of a calorie deficit (net amount of calories burned vs. calories consumed) does he need each day to reach his goal? | 0.0000 | 0.0268 | 0.6406 |

