# OpenReview forum: "Learning Compact Representations of LLM Abilities via Item Response Theory"
_ICLR.cc/2026/Conference — Submitted to ICLR 2026_

### Official Review · Reviewer_dWCK · 2025-10-21

**Soundness:** 2
**Presentation:** 3
**Contribution:** 2
**Rating:** 2
**Confidence:** 3

**Summary:**

The paper introduces a neural network *IrtNet* that is used to estimate an LLM's ability and characteristics of an LLM query. The model is then fit on a portion of embedLLM data for model routing, benchmark prediction, ablation and interpretability experiments.

**Strengths:**

* The authors perform an ablation study demonstrating that their MOE architecture provides improvements over a vanilla NN.
* The OOD prediction experiment demonstrates that IrtNet offers some robustness compared to EmbedLLM (though this should also be compared to other scaling laws see weaknesses)
* The difficulty characteristic and LLM embeddings seem meaningful

**Weaknesses:**

* Insufficient routing evaluation: In the routing literature, a router is generally evaluated on how it performs at selecting models with a performance-cost trade off in mind. The authors opt to test routing capability with performance prediction only. This leaves out a crucial piece of routing, and makes this task hard to distinguish from bench mark prediction. The authors should test their router on a dataset with cost and accuracy (see CARROT or Routerbench if embedLLM lacks this) and provide Area-Under the (Routing) Curve results.
* Lack of Scaling law baselines in benchmark prediction: Prior observational scaling laws ([3], [4]) can be used for benchmark prediction from training data. The authors cite these works but compare their scaling law to none of these, which is a major weakness in the experimental design. Furthermore, ([3], [4]) are substantially more interpretable, so performance gains really need to be significant.
* Lack of clarity in evidence for interpretability: The authors present a t-SNE projection of the query embeddings produced by IrtNet, showing that clustering by benchmark is present. Prior to passing through IrtNet, queries are fed through a pre-trained embedder. I suspect these raw embeddings would exhibit the same clustering behavior, so its unclear to me that IrtNet adds much.

**Questions:**

* What do the t-SNE projections of the raw embeddings look like?
* How does benchmark prediction performance compare to Sloth and other observational scaling laws?
* How does the router perform in temrs of Area Under the (routing) Curve?
* Is possible to verify the meaningfulness of the distinguishability characteristic $\alpha_q$?

---

> ### Author Response · Authors · 2025-11-13
> **Confusing parts of the reviewer comments**
>
> Before starting the rebuttal, we would like to clarify the confusing parts of the reviewer comments.
>
> **W2. Lack of Scaling law baselines in benchmark prediction: Prior observational scaling laws ([3], [4])...**
>
> In our paper, [3], [4] refer to the following paper:
>
> [3] Antoine Bordes, Nicolas Usunier, Alberto Garcia-Duran, Jason Weston, and Oksana Yakhnenko.
> Translating embeddings for modeling multi-relational data. Advances in neural information processing systems, 26, 2013.
>
> [4 ]Li Cai, Kilchan Choi, Mark Hansen, and Lauren Harrell. Item response theory. volume 3, pp.
> 297–321. Annual Reviews, 2016.
>
>  It is obvious that these have nothing to do with scaling laws, so it is quite confusing what specific works the reviewers refer to here as [3] and [4]. Could you please clarify this?
>
> **Q2. How does benchmark prediction performance compare to Sloth and other observational scaling laws?**
>
> We are not entirely clear about which specific works the reviewer is referring to regarding "Sloth" or "other observational scaling laws." If you could provide concrete citations, we would be happy to run the corresponding experiments.

---

> > ### Comment · Reviewer_dWCK · 2025-11-18
> > **Review clarification**
> >
> > Apologies for the missing references. [3], [4] were meant to refer to
> >
> > Polo et al., 2024 (SLOTH)
> >
> > Ruan et al. https://arxiv.org/abs/2405.10938. This is another observational scaing law.
> >
> > The authors could also compare to Zhang et al. https://arxiv.org/abs/2506.07673 which studies benchmark prediction

---

> ### Author Response · Authors · 2025-11-23
> **To Reviewer dWCK**
>
> **R1. The authors should test their router on a dataset with cost and accuracy (see CARROT or Routerbench if embedLLM lacks this) and provide Area-Under the (Routing) Curve results. / How does the router perform in temrs of Area Under the (routing) Curve?**
>
>
> It is unnecessary and unfair to compare our method to cost/quality trade-offs work like CARROT or Routerbench, because they address a different topic and belong to a separate line of research. Our study, by contrast, targets routing purely to boost performance. Indeed, in SOTA research (e.g., RouerDC (NeurIPS'24), EmbedLLM (ICLR'25) and MODEL-SAT (AAAI'25)), these two distinct lines of studies are NEVER compared in experiments.
>
> **R2. What do the t-SNE projections of the raw embeddings look like?/Is possible to verify the meaningfulness of the distinguishability characteristic?**
>
> We have added the t-SNE projections of the raw query embeddings in **Appendix A.5.1**. We conducted t-SNE visualization on $\mathbf{v}_q$ and compared it to the space of learned discrimination vectors $\alpha_q$ (Figure 4 vs. new supplementary Figure 7). While $\mathbf{v}_q$ exhibits some general semantic clustering, the clusters are often diffuse and overlapping. Conversely, the learned $\alpha_q$ space displays significantly tighter, more discrete clustering segregated by benchmark. This confirms that our MoE network acts as a powerful structural filter, transforming generic semantic information ($\mathbf{v}_q$) into a highly distinguishable and interpretable ability-requirement space ($\alpha_q$), which is the core reason for our model's robust performance.
>
> **R3. How does benchmark prediction performance compare to Sloth and other observational scaling laws?**
>
> We appreciate the reviewer's suggestion. However, a direct performance comparison is not feasible because Sloth/OSLs fundamentally differ from IrtNet in prediction granularity and input requirements, rendering them inapplicable to the task setting of this paper.
>
> 1.	Mismatch in Prediction Granularity: While our task involves predicting benchmark scores, it necessitates query-level predictions prior to calculating the average. Sloth/OSLs, in contrast, are benchmark-level predictors. They utilize computational resource parameters ($s, t$) to predict a model's average accuracy across an entire benchmark. Crucially, they lack query awareness—for any two distinct queries $q_1$ and $q_2$ within the same benchmark (e.g., GSM8K), Sloth yields identical predicted scores for the same model.
>
> 2.	Incompatibility of Training Data: The training of Sloth strictly depends on the model's computational metadata (parameter count $s$ and training token count $t$) to fit scaling laws. Conversely, our setting relies exclusively on query semantics and historical performance data; thus, we cannot train Sloth for prediction within our task framework.
>
> Nevertheless, we recognize observational scaling laws, represented by Sloth, as a significant alternative approach for benchmark prediction. Accordingly, we have added a new discussion of these works to the **Related Work** section.

---

> > ### Comment · Reviewer_dWCK · 2025-11-25
> > **rebuttal response**
> >
> > Thanks for adding the raw query embeddings, I agree they clarify IrtNets contribution. That said, I don't agree with the authors on routing or the lack of baselines on benchmark predictions. In my mind the intended contribution of the paper is performance on each of these tasks is the main contribution of the paper.
> >
> > *Routing*: I disagree that routing to boost performance and routing to boost cost efficiency are a 'separate line of research'. While some papers do only look at performance; this can be gamed by picking the best model more often and I do not think I study that looks at cost (the authors could even just use a constant cost predictor) is out of the scope of the work.
> >
> > *Benchmark prediction:* I agree that the inputs to *IrtNet* are not the same as other scaling laws like Sloth, but per query performance predictions can easily be aggregated into benchmark predictions. The authors use open source models where parameters and model counts are  (mostly) available. Even if the authors do not want to compare to scaling laws, there seem to be many other benchmark prediction methods (https://arxiv.org/pdf/2506.07673 covers some) that the authors do not compare to. This contrbution is not very convincing with only two baselines.

---

> > > ### Author Response · Authors · 2025-11-27
> > > **To Reviewer dWCK**
> > >
> > > **Benchmark Prediction Baselines**
> > >
> > > We have expanded our evaluation to include Observational Scaling Laws (OSL) [Ruan et al., 2024] and Random-Sampling-Learn (RSL) [Zhang et al., 2025] as additional baselines in **Appendix A.3.1**, which can be seen in the latest revised pdf. Note that we focuse on these two because Sloth[Polo et al., 2024] relies on proprietary training token counts, which are unavailable for most models in our dataset, making a fair reproduction infeasible.
> > >
> > > We maintain the Out-of-Distribution (OOD) prediction setup. To enable OSL and RSL, we grant them additional supervision: reserving 5% of models (with 50 cold-start queries + ground truth scores) for training their regressors. In contrast, IrtNet requires no such historical model data for the target benchmark. Despite this advantage for the baselines, IrtNet significantly outperforms them:
> > >
> > > IrtNet achieves the lowest average RMSE of 0.16.
> > >
> > > OSL and RSL lag behind with average RMSEs of 0.32 and 0.29, respectively.
> > >
> > > The results confirm our hypothesis regarding prediction granularity. OSL and RSL operate at a benchmark-level, inferring aggregate scores based on historical trends, which struggles to generalize to unseen OOD benchmarks. IrtNet’s superior performance stems from its fine-grained, query-level mechanism, which leverages semantic understanding to build precise bottom-up predictions, robustly capturing capability requirements even in zero-shot scenarios.
> > >
> > > **Routing**
> > >
> > > We believe our current evaluation is rigorous and sufficient for the following reasons:
> > >
> > > 1. **Established Research Scope & Precedents:** We have shown that state-of-the-art routing studies, such as RouterDC (NeurIPS'24), EmbedLLM (ICLR'25), and MODEL-SAT (AAAI'25), focus exclusively on maximizing routing accuracy. Following this established standard, these works do not compare against budget-constrained baselines.
> > >
> > > 2.  **Foundation of Routing:** Performance-cost trade-off  routing can be viewed as a multi-objective optimization problem balancing **performance** and **cost**. Our work focuses on the most fundamental and challenging component: **accurately estimating the probability of correctness**, $P(y=1|m, q)$. As noted in the cited literature, cost-aware routing is essentially a weighted function of this probability and cost (e.g., $Score = P - \lambda \cdot Cost$). Without accurate performance estimation ($P$), no cost-tradeoff can be effective. IrtNet provides this critical foundation.
> > >
> > > 3.  **No "Gaming" by Size:** The concern that performance-only routing might "game" the metric by selecting the largest models does not apply to our experimental setting. As detailed in the **Appendix A.2.1**, our candidate pool largely consists of models with **similar parameter scales (7B-9B) and performances**. In this homogeneous setting where simply "picking the best" is not a viable strategy, IrtNet still achieves the best **Micro-Average accuracy**. This proves that our gains stem from genuine semantic understanding of query-model compatibility, not from exploiting model size.
> > >
> > > 4.  **Data Constraint, Not Method Constraint:** The omission of cost-aware experiments is strictly due to **data limitations**, not algorithmic ones. The public datasets used in our study lack the necessary inference metadata (e.g., generation token counts or API pricing) to construct a realistic cost function. However, our architecture is inherently flexible and can seamlessly support cost-aware routing objectives whenever such metadata is available.

---

### Official Review · Reviewer_iprt · 2025-10-30

**Soundness:** 2
**Presentation:** 2
**Contribution:** 2
**Rating:** 4
**Confidence:** 4

**Summary:**

This paper proposes a novel framework for learning compact representations of LLM abilities using Item Response Theory (IRT). The authors introduce IrtNet, a (MoE) network designed to jointly learn query characteristics such as difficulty and discrimination. The proposed method demonstrates strong performance on downstream tasks, including model routing and benchmark accuracy prediction.

**Strengths:**

The grounding of the capability modeling process in Item Response Theory provides a novel perspective by incorporating the difficulty and discrimination of queries.

The framework's effectiveness is demonstrated by its strong performance on downstream tasks like model routing and benchmark accuracy prediction.

**Weaknesses:**

The motivation for employing both Item Response Theory and the MoE network architecture is not fully elaborated. The paper would benefit from further discussion and ablation studies comparing the proposed item response function with simpler alternatives, such as plain logistic regression.

The applicability of the framework may be limited, as the evaluation is restricted to models from the EmbedLLM dataset. The paper does not explore whether this method can be generalized to more recent and capable models. Furthermore, the specific models used in the model routing task should be explicitly stated.

The primary novelty compared to EmbedLLM's encoder-decoder architecture is the introduction of the IRT formalism and the IrtNet architecture. However, the paper does not sufficiently justify that the source of the advantages of this new formulation over the previous approach lies in the IRT formalism or the MoE architecture.

**Questions:**

The formulation used in the paper appears to be a bit different from the standard 2PL model, where the discrimination parameter typically interacts multiplicatively with both the ability and difficulty parameters. Could the authors clarify the rationale behind their proposed formulation?

---

> ### Author Response · Authors · 2025-11-23
> **To Reviewer iprt**
>
> **R1. The motivation for employing both Item Response Theory and the MoE network architecture is not fully elaborated**
>
>
> Inspired by psychometrics (such as the GRE exam), we introduce IRT (Item Response Theory) with the aim of explicitly disentangling three latent variables mathematically: model capability, query difficulty, and query discrimination. It is this theory-based disentanglement that enables IrtNet not only to achieve more accurate out-of-distribution (OOD) prediction through difficulty awareness but also to provide interpretability that traditional black-box models cannot offer.
>
> **R2. The paper would benefit from further discussion and ablation studies comparing the proposed item response function with simpler alternatives, such as plain logistic regression.**
>
> We update our ablation study experiment in **Table 4**. After removing the MoE layer or IRT formulation, IrtNet exhibits a significant performance drop in the model routing task and a slight decline in the correctness prediction task.  This ablation result fully demonstrates the effectiveness of the MoE structure and the IRT formulation.
>
> **R3. The applicability of the framework may be limited, as the evaluation is restricted to models from the EmbedLLM dataset. / The paper does not explore whether this method can be generalized to more recent and capable models.**
>
> We have added new dataset experiments in **Appendix A.2.1**. We select 10 datasets adopted by Avengers and more recent LLMs to perform routing experiments, and still achieve SOTA results.
>
> **R4. the specific models used in the model routing task should be explicitly stated.**
>
> We have added the specific list of models included in the **Appendix A.6**.
>
> **R5. However, the paper does not sufficiently justify that the source of the advantages of this new formulation over the previous approach lies in the IRT formalism or the MoE architecture.**
>
> See **R2**.
>
> **R6. The formulation used in the paper appears to be a bit different from the standard 2PL model. Could the authors clarify the rationale behind their proposed formulation**
>
> Our formulation is mathematically equivalent slope-intercept form of the 2PL model (where $\beta_{ours} = a \cdot b_{standard}$). We chose this parameterization to avoid the multiplicative coupling between discrimination and difficulty found in the standard formulation. Decoupling them into a dot product and a bias term ($\alpha^\top \theta - \beta$) significantly improves numerical stability and gradient flow during end-to-end neural network training.

---

> > ### Comment · Reviewer_iprt · 2025-11-26
> >
> > Thank you for your response. The result presented in Table 4 and the Appendix are nice, and an additional suggestion is to include the baseline performance (such as EmbedLLM) in the table for easier comparison.
> >
> > However, the motivation for using MoE is still not sufficiently clear. The core of your explanation should be a clear motivation explaining why the MoE architecture was chosen and what its specific advantages are in this context, rather than simply stating that the performance improved slightly after switching to the MoE architecture. If this point can be adequately addressed, I will consider adjust my score.

---

> > > ### Author Response · Authors · 2025-11-26
> > > **To Reviewer iprt**
> > >
> > > **Include the baseline performance (such as EmbedLLM)**
> > >
> > > In fact, following IrtNet without MoE and Irt is the matrix factorization method of EmbedLLM (plain logistic regression), and we will add an explanation of this point in the next version update.
> > >
> > > **Explain why the MoE architecture was chosen**
> > >
> > > We acknowledge that our initial explanation was result-oriented. The choice of the MoE architecture was driven by three fundamental motivations rooted in the nature of the LLM evaluation task:
> > >
> > > 1. Handling Semantic Heterogeneity: The input queries originate from highly diverse domains (e.g., Mathematics, Coding, Medicine, Commonsense). A standard MLP suffers from negative interference when trying to learn a universal mapping for such disjoint semantic distributions. The MoE architecture allows the network to decouple these domains, enabling different experts to specialize in extracting difficulty features from different semantic subspaces, thereby mitigating interference.
> > >
> > > 2. Modeling Compositional Ability Requirements: The discrimination vector $\alpha_q$ represents the specific "ability profile" required to solve a query. Many complex queries require a composition of skills (e.g., a math word problem requires both reasoning and reading comprehension). The MoE architecture naturally supports this via its gating mechanism, which learns to dynamically weigh and combine the outputs of specialized experts, enabling a flexible, compositional representation that a static MLP cannot easily capture.
> > >
> > > 3. Scalability for Future Pre-training: We designed IrtNet with future scalability in mind. Just as text embedding models (e.g., Qwen-Embedding) scale up to handle massive corpora, we envision IrtNet scaling to become a universal difficulty evaluator pre-trained on millions of diverse queries. The MoE architecture is uniquely suited for this trajectory: it allows us to significantly increase the model's parameters to encode vast amounts of domain-specific knowledge while maintaining efficient inference speeds. This architectural choice ensures that IrtNet can scale gracefully as the volume and diversity of LLM evaluation data explode.
> > >
> > > In summary, MoE is not just a performance booster but a structural necessity for handling the multi-domain, compositional nature of heterogeneous LLM benchmarks.

---

> > > > ### Comment · Reviewer_iprt · 2025-11-27
> > > >
> > > > Thank you for your response. Now the motivation is much clear, and I have adjusted my score.

---

### Official Review · Reviewer_iCAe · 2025-10-30

**Soundness:** 3
**Presentation:** 3
**Contribution:** 3
**Rating:** 4
**Confidence:** 3

**Summary:**

This paper proposes IRTNet, a novel framework for learning compact and interpretable representations of large language model (LLM) abilities, inspired by Item Response Theory (IRT). This new framework can estimate the probability that a specific model will answer a specific query correctly. With a MoE network for parameter coupling, this model learns model's ability embedding, query discrimination vector and difficulty scalar jointly. The experiments shows the outperformance of the new method in downstream tasks, such as model routing and benchmark prediction.

**Strengths:**

1. The methodology is described clearly and easy for the readers to follow.
2. The application of IRT model in LLM embedding learning is a novel idea.
3. The comprehensive experiments cover most downstream tasks with LLM representations, and the outperformance support the efficacy of IrtNet.

**Weaknesses:**

1. Compared to EmbedLLM, which also tries to construct compact representations of LLMs' abilities, the new method proposed in this paper applies IRT to modeling and introduces ab MoE-based method. Although the experiments show IrtNet's outperformance, the motivation for designing is not fully illustrated in the paper. I think a paragraph to analyze why IRT and query embeddings can achieve the gain will be very insightful.
2. For validating difficulty in Section 5.2, the results is still on the level of benchmark. I think the difficulty parameter is actually on the level of query. It would be very interesting to show some query-level results. Actually there is a previous benchmark named Easy2Hard-Bench [1]. They also applied IRT method on the model's answer correctness on different queries, but their goal is to estimate the difficulty of the queries. Their published datasets include GSM8K, which is also in the datasets here. You could compare the beta parameters in IrtNet and the difficulty scores from Easy2Hard-Bench.
3. You could also try to do some human evaluations to compare the difficulty estimated by IrtNet and human's judgement, which is also applied in Easy2Hard-Bench [1]. Although it may not be relevant to the main storyline in the paper, this comparison would provide another perspective to understand LLM's abilities via human perspective.

I would like to adjust my score if the authors can address these questions.

[1] Ding, Mucong, et al. "Easy2Hard-Bench: Standardized Difficulty Labels for Profiling LLM Performance and Generalization." Advances in Neural Information Processing Systems 37 (2024): 44323-44365.

**Questions:**

1. The results of benchmarking prediction shown in Figure 3 indicates that both IrtNet and EmbedLLM tend to overestimate the LLMs' performance on benchmarking. Could you please accordingly provide some insights?
2. There are four variants of IRT model (1PL-4PL). What the authors used in IrtNet is more like to 2PL. Could you please explain how you decide to use this?
3. Does IrtNet have some potential limitations? It would be good to discuss in the paper.

---

> ### Author Response · Authors · 2025-11-23
> **To Reviewer iCAe**
>
> **R1. Although the experiments show IrtNet's outperformance, the motivation for designing is not fully illustrated in the paper.**
>
> Same as Reviewer iprt **R1**.
>
> We have elaborated the motivation from lines xxx to xxxx in our initial submission. In general, Inspired by psychometrics (such as the GRE exam), we introduce IRT (Item Response Theory) with the aim of explicitly disentangling three latent variables mathematically: model capability, query difficulty, and query discrimination. It is this theory-based disentanglement that enables IrtNet not only to achieve more accurate out-of-distribution (OOD) prediction through difficulty awareness but also to provide interpretability that traditional black-box models cannot offer.
> We are happy to clarify further if the reviewer can specify which parts were not fully explained.
>
> **R2. It would be very interesting to show some query-level results. You could compare the beta parameters in IrtNet and the difficulty scores from Easy2Hard-Bench.**
>
> We have added the presentation of query-level results and their alignment with the Easy2Hard-Bench difficulty in **Appendix A.5.2**. Overall, our $\beta$ parameter shows a good correlation with the Easy2Hard-Bench, once again demonstrating the strong interpretability of the $\beta$ parameter.
>
> **R3. The results of benchmarking prediction shown in Figure 3 indicates that both IrtNet and EmbedLLM tend to overestimate the LLMs' performance on benchmarking. Could you please accordingly provide some insights?**
>
> We have supplemented the presentation of prediction results for all datasets in **Appendix A.3**. The prediction bias includes both overestimation and underestimation, which mainly depends on the distribution of the training set itself and the convergence of the model training. Overall, the patterns of overestimation or underestimation observed in IrtNet are consistent with those in Embedllm.
>
> **R4. There are four variants of IRT model (1PL-4PL). What the authors used in IrtNet is more like to 2PL. Could you please explain how you decide to use this?**
>
> We chose the 2PL model because it strikes the optimal balance between expressiveness and learnability for LLM evaluation. Necessity of Discrimination (vs. 1PL): Unlike 1PL, 2PL captures the varying discriminative power of queries ($\alpha_q$). This is critical because complex reasoning tasks distinguish strong models much better than simple fact-retrieval tasks, a nuance 1PL fails to capture. Stability & Efficiency (vs. 3PL/4PL): While 3PL/4PL model guessing/slipping, they introduce significant optimization instability when jointly trained with high-dimensional embeddings. For our core tasks—routing and ranking—2PL provides sufficiently accurate probability estimates without the optimization overhead and overfitting risks of higher-order models."
>
> **R5. Does IrtNet have some potential limitations?**
>
> I believe the biggest limitation of IrtNet (consistent with most other mainstream routers) is its reliance on the Query Encoders as input. Because the query is transformed into a single embedding vector, some semantic information is inevitably lost compared to the raw tokens, which ultimately limits the upper bound of this category of methods.

---

### Official Review · Reviewer_c1ve · 2025-11-01

**Soundness:** 3
**Presentation:** 3
**Contribution:** 3
**Rating:** 4
**Confidence:** 3

**Summary:**

The paper proposes IrtNet, a framework that learns compact, interpretable representations of LLM abilities for two downstream tasks: model routing and benchmark prediction. It casts the probability that a model answers a query correctly as a 2-parameter IRT response. Query characteristics are produced from a dense MoE over query embeddings; model embedding is learned jointly, end-to-end, by minimizing BCE on model–query correctness labels. Experiments on 10 benchmarks (35k queries, 112 models) report state-of-the-art routing accuracy and data-efficient benchmark prediction; the authors also present interpretability analyses (difficulty correlates with empirical hardness; embeddings cluster by family/specialization).

**Strengths:**

1. Uses a dense MoE to map query embeddings to IRT parameters, arguing better stability vs sparse MoE and improved discrimination estimates—novel architectural choice for this setting.
2. Strong empirical results: IrtNet achieves the best micro/macro routing accuracy across 10 datasets;
3. Addresses a practical need: scalable, cost-aware LLM orchestration via interpretable ability representations; potentially impactful for evaluation, routing, and system design at scale.

**Weaknesses:**

1. While routing baselines are strong, the paper doesn’t compare with cascaded inference or test-time adaptation strategies that may implicitly capture difficulty (e.g., multi-try reasoning depth) and could be competitive on cost/quality trade-offs. Please position IrtNet against those classes more explicitly and, if feasible, add at least one cascade baseline.
2. OOD experiment is not very promising. They are also not added to the main manuscript but are present in the appendix.
3. Although the paper targets routing at scale, it lacks end-to-end latency/throughput analyses.
4. The method relies on a fixed text embedding model (all-mpnet-base-v2) to seed the MoE. There’s no sensitivity analysis to the embedding choice/domain shift.

**Questions:**

Check weakness above

1. MoE design choices
Why dense MoE over sparse MoE—besides stability? Did you experiment with different expert counts and routing strategies? An expert-count sweep and a sparse-vs-dense comparison (equal parameters) would contextualize the ablation.

---

> ### Author Response · Authors · 2025-11-13
> **Confusing parts of the reviewer comments**
>
> Before starting the rebuttal, we would like to clarify the confusing parts of the reviewer comments.
>
> **The paper doesn’t compare with cascaded inference or test-time adaptation strategies**
>
> We are not entirely clear about which specific works the reviewer is referring to regarding "cascaded inference" or "test-time adaptation strategies." If you could provide concrete citations, we would be happy to run the corresponding experiments.

---

> > ### Comment · Area_Chair_vcM2 · 2025-11-18
> > **Take a look at this request**
> >
> > Can you please clarify what you think would be the relevant additions?

---

> ### Author Response · Authors · 2025-11-23
> **To Reviewer c1ve:**
>
> **R1.the paper doesn’t compare with cascaded inference or test-time adaptation strategies that may implicitly capture difficulty (e.g., multi-try reasoning depth)**
>
> We would appreciate clarification from the reviewer regarding the intended meaning of ‘cascaded inference’ or ‘test-time adaptation.’ As the reviewer has not yet responded, we will defer a detailed reply to this point.”
> Cascaded inference or test-time adaptation strategies improve performance through rollouts at test time (such as multiple sampling), which is fundamentally different from the large category of router methods discussed in this paper. IrtNet-like router methods only require a single routing decision before selecting one LLM to call. In contrast, cascaded inference may require calling one or more LLMs multiple times. This leads to cascaded inference having a much greater computational overhead and latency during actual inference compared to traditional routers. Therefore, we believe that it is not appropriate to compare these two classes of methods directly.
>
> **R2. Be competitive on cost/quality trade-offs**
>
> It is unnecessary and unfair to compare our method to cost/quality trade-offs work like CARROT or Routerbench, because cost/quality trade-offs typically belong to a separate line of research. Our study, by contrast, targets routing purely to boost performance. Note that in SOTA research (e.g., RouerDC (NeurIPS'24), EmbedLLM (ICLR'25) and MODEL-SAT (AAAI'25)), these two distinct lines of studies are NEVER compared in experiments.
>
> **R3. OOD experiment is not very promising. They are also not added to the main manuscript but are present in the appendix.**
>
> The OOD setting is generally the most difficult setting because if a model cannot relate a test case to past experience, it essentially has to guess. Consequently, the gap between our performance and the embedding-based methods is smaller in this setting. However, we still maintain a lead. Additionally, the OOD results were indeed presented in the main manuscript **Section 4.3** of the original version.
>
> **R4. Although the paper targets routing at scale, it lacks end-to-end latency/throughput analyses.**
>
> We have added a section on computational cost in **Appendix A.1**. In short, to process a problem and select the best model from 112 candidates, IrtNet only takes 6.7 milliseconds on average. Our method is extremely efficient. With a batch size of 112 (corresponding to the candidate model pool size), determining the best model for a query takes only ~6.7 ms on average in our experiments (20 seconds for 3,000 queries).
>
> **R5. The method relies on a fixed text embedding model (all-mpnet-base-v2) to seed the MoE. There’s no sensitivity analysis to the embedding choice/domain shift.**
>
> We have included ablation study on query encoders in **Appendix A.4.1**. Even when switching to inferior encoders, we still maintain an exceptionally strong performance level. Furthermore, regardless of which encoder is used, we ensure that all methods use the same encoder, and our method does not benefit more from it than others.
>
> **R6. MoE design choices Why dense MoE over sparse MoE—besides stability? Did you experiment with different expert counts and routing strategies? An expert-count sweep and a sparse-vs-dense comparison (equal parameters) would contextualize the ablation.**
>
> The primary reason for using dense MoE (Mixture-of-Experts) is purely for stability, and for no other reason. This is determined by the relatively small amount of training data we have. In **Appendix A.4.3**, we have added a comprehensive ablation study varying both total experts and active experts count while keeping total parameters constant. We compare a Dense MoE (N=40, K=40) against a Sparse MoE (N=40, K=4/20). Due to our limited data size, the sparse MoE indeed suffered from insufficient training, and the model’s performance gradually improved as the number of activated experts
> increased. In the dense setting, the model’s performance change was largely insignificant.

---

### Official Review · Reviewer_r2sv · 2025-11-01

**Soundness:** 3
**Presentation:** 3
**Contribution:** 3
**Rating:** 6
**Confidence:** 3

**Summary:**

The paper presents IrtNet, a neural adaptation of item response theory that embeds large language models and benchmark queries into shared compact representations for predicting correctness, routing tasks, and benchmarking. By combining model embeddings with query-specific discrimination vectors and difficulty scalars produced by a dense mixture-of-experts layer, the method models the probability of an LLM answering a query correctly and demonstrates state-of-the-art results on the EmbedLLM dataset for model routing accuracy, benchmark score prediction, and data-efficient correctness estimation.

**Strengths:**

The formulation tightly couples psychometric intuition with neural modeling, yielding a simple response function while learning expressive query parameters via the MoE. Experiments cover key downstream tasks, routing, in-distribution prediction at varying data scales, and leave-one-benchmark-out generalization, and show consistent gains over strong baselines such as EmbedLLM and Avengers-Pro. The analysis section goes beyond raw metrics, validating that learned difficulty correlates with empirical benchmark hardness, visualizing discrimination vectors that cluster by dataset, and probing LLM embeddings for family and specialization structure. These pieces collectively support both effectiveness and interpretability claims.

**Weaknesses:**

The evaluation is restricted to the EmbedLLM dataset with majority-voted correctness labels, so transfer to free-form generation, human preference judgments, or settings with label noise remains untested. Training details such as computation cost, convergence behavior, or sensitivity to the representation dimension and number of experts are sparse, making reproducibility and deployment considerations less clear. Although the architecture is inspired by IRT, comparisons to more recent multi-dimensional or causal IRT baselines are missing, and ablations only test removing the MoE rather than alternative query encoders or different priors on difficulty. The routing evaluation assumes access to oracle correctness labels for supervision, but practical systems may suffer from biased or delayed feedback—this is not discussed.

**Questions:**

How robust is IrtNet to inconsistent or adversarially noisy correctness annotations when learning the model and query embeddings? In routing scenarios with partial feedback, could the authors clarify how the system would update θ and α without exhaustive supervision? What are the computational requirements for training and inference relative to baselines, and do they scale gracefully as the number of models or benchmarks grows? Finally, have the authors explored adapting the framework to non-binary outcomes, such as partial credit or calibrated probability scores, to better align with emerging evaluation datasets?

---

> ### Author Response · Authors · 2025-11-23
> **To Reviewer r2sv:**
>
> **R1. The evaluation is restricted to the EmbedLLM dataset.**
>
> We have added new dataset experiments in **Appendix A.2.1**. We select 10 datasets adopted by Avengers, which is the state-of-the-art routing method [1], and more recent LLMs (such as DeepSeek-R1-Distill-Qwen-7B, Llama-3.1-8B-Instruct, Qwen3-8B, gemma-
> 2-9b-it) to perform routing experiments. Our proposed method still achieve SOTA results.
>
> **R2. Settings with label noise remains untested/How robust is IrtNet to inconsistent or adversarially noisy correctness annotations when learning the model and query embeddings?**
>
> We have added new noisy setting experiments in **Appendix A.2.2**. The results show that IrtNet demonstrates robustness to noisy data and still maintains SOTA results.
>
> **R3. Free-form generation, human preference judgments are not tested/ In routing scenarios with partial feedback, could the authors clarify how the system would update θ and α without exhaustive supervision?**
>
> **Free-form generation**
>
> We prioritized standard benchmarks (e.g., MMLU, GSM8K) to rigorously validate the IRT framework on objective correctness, avoiding the subjectivity of free-form evaluation. However, IrtNet is methodology-agnostic to the task type. For open-ended tasks, we can leverage LLM-as-a-Judge (e.g., GPT-4 grading) or Reward Models to generate numerical scores (e.g., binary acceptance or 1-5 quality ratings) as labels. Once these proxy labels are obtained, IrtNet can model them exactly as it does for benchmark accuracy.
>
> **Updating with Partial Feedback**
>
> During offline training, we only need to collect those examples where the query, the model, and the label are complete to ensure that $\theta$ and $\alpha$ are updated through backpropagation.
>
>
> **R4. Training details such as computation cost, convergence behavior, or sensitivity to the representation dimension and number of experts are sparse, making reproducibility and deployment considerations less clear.**
>
> We have included more detailed experimental specifics in **Appendix A.1**, covering aspects such as memory consumption, training and testing time, and number of training epochs. Simply put, our method has very low memory consumption and is fast to run.
>
> Additionally, we have added ablation study on the representation dimension and number of experts in **Appendix A.4**, which demonstrates the robustness of IrtNet.
>
> Moreover, for reproducibility, note that we have already released our code and adopted the same settings as in previous studies (EmbedLLM).
>
> **R5. Although the architecture is inspired by IRT, comparisons to more recent multi-dimensional or causal IRT baselines are missing**
>
> To the best of our knowledge, there is currently no work that utilizes IRT to learn a unified model representation for downstream tasks.
>
> **R6. Ablations only test removing the MoE rather than alternative query encoders or different priors on difficulty.**
>
> In our initial submission, we have already included ablation study on query encoders in **Appendix A.4.1**. Even when switching to inferior encoders, we still maintain an exceptionally strong performance level. Furthermore, regardless of which encoder is used, we ensure that all methods use the same encoder, and our method does not benefit more from it than others.
>
> **R7. What are the computational requirements for training and inference relative to baselines, and do they scale gracefully as the number of models or benchmarks grows?**
>
> Regarding VRAM requirements (See **R4**).
>
> When the number of models increases, the only parameters IrtNet needs to learn that grow linearly are the model embeddings. Crucially, these parameters are very few compared to the total number of parameters (each model is a 232-dimensional vector, so 100 models only account for roughly 23,200 parameters). Therefore, the computational overhead is generally manageable. When the benchmarks grow, only the training time increases linearly (because each sample must participate in the loss calculation), but the number of parameters IrtNet needs to learn remains constant. This is because dimensions of discrimination $\alpha_q$ and the difficulty $\beta_q$ do not change as the data increases, thus the computational overhead during inference is not increased.
>
> **R8. Finally, have the authors explored adapting the framework to non-binary outcomes, such as partial credit or calibrated probability scores, to better align with emerging evaluation datasets?**
>
> We agree that non-binary outcomes would be an interesting venue to explore as future work. In this paper, we prioritized standard benchmarks (e.g., MMLU, GSM8K) to rigorously validate the IRT framework on objective correctness, avoiding the subjectivity of free-form evaluation. We believe that this type of scenario is highly valuable and also full of challenges, and therefore, we would be very happy to explore the non-binary outcomes setting in our next work.

---

### Author Response · Authors · 2025-12-02
**Brief summary of the rebuttal**

We sincerely thank all reviewers for their thorough evaluation, constructive feedback, and the time they invested in improving our work. We carefully considered every comment and revised the paper accordingly. Below, we summarize the main concerns raised by each reviewer and how they have been addressed in the updated manuscript.

## **Reviewer r2sv**

The reviewer noted that the evaluation was limited, robustness under noisy or partial feedback was missing, and important experimental details (computational cost, convergence, sensitivity) were insufficient.

To address this, we expanded the empirical scope by adding **new datasets and recent LLMs in Appendix A.2.1**, incorporated **noisy-label robustness experiments in Appendix A.2.2**, and added detailed **training-time, memory, runtime, and convergence statistics in Appendix A.1**. We also included **representation-dimension and expert-count sensitivity studies in Appendix A.4**, and clarified how the method generalizes to free-form and partial-feedback tasks in the updated explanations within the paper’s methodology section.

## **Reviewer c1ve**

This reviewer requested clarification regarding missing comparisons to cascaded/test-time adaptive strategies, clearer OOD presentation, and additional analysis of latency, encoder sensitivity, and MoE design.

We distinguished router-style methods from cascaded inference in the updated discussion in **rebuttal**, and emphasized that OOD results are already presented in **Section 4.3**. We added **end-to-end latency and throughput measurements in Appendix A.1**, and extended robustness analysis with **query-encoder ablations in Appendix A.4.1**. To further clarify design choices, we added **dense vs. sparse MoE experiments and discussion in Appendix A.4.3**.

## **Reviewer iCAe**

The reviewer requested clearer motivation for combining IRT with query embeddings, query-level validation of difficulty, more insight into prediction bias, justification for the 2PL choice, and an explicit limitations discussion.

We expanded the motivation discussion in the main manuscript. We added **query-level difficulty validation and alignment with Easy2Hard-Bench in Appendix A.5.2**, and included complete prediction-bias visualizations for all datasets in **Appendix A.3**. We clarified the choice of **2PL** within **rebuttal** and provide an **limitations discussion**.

The reviewer stated that he/she would like to adjust his/her score.

## **Reviewer iprt**

This reviewer questioned the necessity of both IRT and MoE, suggested comparing with simpler formulations, asked for clearer demonstration of generalization beyond EmbedLLM models, and requested clarification of the IRT parameterization.

We added **ablations removing IRT or the MoE layer in Table 4**, showing clear performance drops without these components. We expanded the evaluation to include **new datasets and updated LLM lists in Appendix A.2.1 and Appendix A.6**, demonstrating generalization to more capable models. Additionally, we clarified that our formulation corresponds to a **slope–intercept equivalent of the standard 2PL model** in **rebuttal**.

The reviewer later stated that the motivation was clear and **updated his/her score**.

## **Reviewer dWCK**

The reviewer emphasized the absence of cost–accuracy trade-off routing, missing scaling-law baselines for benchmark prediction, and requested stronger evidence that the learned representations provide additional structure.

We clarified the scope distinction in **rebuttal**, explaining why cost-aware routing is treated as a separate line of work in prior literature. We added **OSL and RSL benchmark-prediction baselines in Appendix A.3.1**, evaluated under the same OOD setup. Additionally, we strengthened interpretability evidence by adding **t-SNE comparisons of raw query embeddings vs. learned discrimination vectors in Appendix A.5.1**, which demonstrate substantially clearer clustering in the learned space.

---

### Meta-Review · Area_Chair_LNBL · 2025-12-27

**Summary:**

The authors propose to learn an MoE-based item response theory (IRT) model that can take in an input query and return a multi-dimensional discrimination vector and a difficulty scalar. This is a straightforward application of IRT; the authors show that it helps with model routing and with predicting whether a given model will be correct on a new query. Reviewers were generally negative (scores before rebuttal were 6 4 4 4 2) and asked questions about experimental comparisons and novelty. The authors generally present their work as the first to use IRT in the context of LLMs, which is not 100% correct, although many similar works fall just within the 4-month concurrent work boundary; they do compare with tinyBenchmarks, a 2024 paper that uses IRT to do benchmark subset selection, and it is unclear why their framing of benchmark prediction is better/more useful than what's in tinyBenchmarks.

Overall, given the generally negative sentiment of the reviewers (though a couple of them indicated that they would increase their score post-rebuttal) and the amount of changes in the rebuttal, I recommend that the authors revise their submission in accordance with the reviews, provide a more thorough comparison with prior work, and resubmit.

**Reviewer Concerns:**

See main comment.

**Reviewer Scores:**

See main comment. I expect two reviewers would have increased their score by 1 or 2.

---

### Decision · Program_Chairs · 2026-01-26

Reject